# Exploring T-Cell Immunity to Hepatitis C Virus: Insights from Different Vaccine and Antigen Presentation Strategies

**DOI:** 10.3390/vaccines12080890

**Published:** 2024-08-06

**Authors:** Gabriel L. Costa, Giuseppe A. Sautto

**Affiliations:** Florida Research and Innovation Center, Cleveland Clinic, Port Saint Lucie, FL 34987, USA; costag2@ccf.org

**Keywords:** Hepatitis C Virus (HCV), vaccine, cellular immunity, T cell

## Abstract

The hepatitis C virus (HCV) is responsible for approximately 50 million infections worldwide. Effective drug treatments while available face access barriers, and vaccine development is hampered by viral hypervariability and immune evasion mechanisms. The CD4+ and CD8+ T-cell responses targeting HCV non-structural (NS) proteins have shown a role in the viral clearance. In this paper, we reviewed the studies exploring the relationship between HCV structural and NS proteins and their effects in contributing to the elicitation of an effective T-cell immune response. The use of different vaccine platforms, such as viral vectors and virus-like particles, underscores their versability and efficacy for vaccine development. Diverse HCV antigens demonstrated immunogenicity, eliciting a robust immune response, positioning them as promising vaccine candidates for protein/peptide-, DNA-, or RNA-based vaccines. Moreover, adjuvant selection plays a pivotal role in modulating the immune response. This review emphasizes the importance of HCV proteins and vaccination strategies in vaccine development. In particular, the NS proteins are the main focus, given their pivotal role in T-cell-mediated immunity and their sequence conservation, making them valuable vaccine targets.

## 1. Introduction

The hepatitis C virus (HCV) represents a significant public health concern worldwide, with an estimated 50 million people living with a chronic infection according to the World Health Organization (WHO) [1]. The virus exhibits considerable genetic diversity, with 8 major genotypes and 86 subtypes identified thus far. Among these, genotype 1 is the most prevalent globally, particularly in North America and Europe, while genotype 3 is common in Southeast Asia and some areas of South Asia [2,3,4]. Knowing the distribution of genotypes is crucial for tailoring treatment and prophylactic strategies and predicting clinical outcomes. Structurally, HCV is a positive-strand RNA virus surrounded by a lipid envelope nucleocapsid and viral envelope-associated proteins. The viral genome encodes structural and non-structural (NS) proteins essential for viral replication and assembly. These include the structural protein core, envelope glycoproteins (E1 and E2), and the NS proteins p7, NS2, NS3, NS4A, NS4B, NS5A, and NS5B [5] (Figure 1). The strategy of targeting these proteins has been the cornerstone of antiviral therapy development.

Historically, the first available treatments for HCV were limited and often associated with significant side effects. Interferon (IFN)-based regimens combined with ribavirin represented the standard of care for many years, but their efficacy was modest and were affected by poor tolerability [6]. Importantly, the landscape of HCV infection treatment has been revolutionized with the introduction of direct-acting antiviral (DAA) drugs. These highly effective, well-tolerated oral medications target specific viral proteins, disrupting viral replication, leading to sustained virologic response rates exceeding 95% in most cases [7]. However, despite the remarkable success of DAAs, several challenges remain in the global effort to control HCV. Access to treatment is a significant barrier in many parts of the world, particularly in low- and middle-income countries, where healthcare resources are limited [8]. Additionally, certain populations, such as injection drug users and incarcerated individuals, face unique challenges in accessing diagnosis and adhering to treatment [9]. Efforts to address these disparities will be essential for achieving the WHO’s goal of eliminating viral hepatitis as a public health threat by 2030 [10].

The immune response to HCV infection is complex and often inadequate for viral clearance. The virus has evolved multiple mechanisms to evade the host immune response, including rapid mutation rates leading to antigenic variability and interference with host immune signaling pathways [11]. Despite the development of adaptive immune responses, many individuals fail to achieve viral clearance and progress to chronic infection, which can lead to liver cirrhosis, hepatocellular carcinoma, and liver failure [12]. The development of an effective HCV vaccine represents a promising strategy for long-term infection control and prevention. However, several obstacles have hampered vaccine development efforts, including the high genetic variability of the virus and the lack of suitable animal models for preclinical testing [13]. Despite these challenges, ongoing research efforts aim to overcome these barriers and develop an effective vaccine that could complement existing treatment strategies in the fight against HCV.

### The HCV Infection and Cell-Mediated Immune Response

The HCV-specific CD4+ and CD8+ T cells play an important role in viral clearance, targeting different viral antigens, with the NS proteins being the major targets [14]. This response is mainly driven by the function of the CD8+ T cells recognizing viral peptides presented on major histocompatibility complex (MHC) class I molecules and directly targeting and killing virus-infected cells. Previous studies, using the chimpanzee model, demonstrated the viremia persistence after CD4+ and CD8+ T-cell depletion [15,16]. Other evidence also supports the importance of the cell-mediated immune response, where both CD4+ and CD8+ are linked to liver inflammation and the correlation between certain human leukocyte antigens (HLAs), class I and II alleles, and HCV clearance or persistence and treatment [17,18]. In fact, the downregulation of the MHC and mutations within CD4+ and CD8+ T-cell HCV epitopes are important factors in HCV-specific T-cell response failure, along with T-cell exhaustion [19,20]. However, in the context of chronic HCV infection, the CD8+ T-cell response is long lasting, while the CD4+ T-cell response wanes over time [21]. Thus, both subsets of cells must be induced, and more importantly, sustained, in order to achieve viral clearance. Therefore, in this scenario, the challenge consists of developing a prophylactic approach capable of overcoming HCV mutations that leads to viral escape from the immune system, and T-cell exhaustion.

As mentioned, T cells play a central role in the immune response to HCV infection. It is essential to understand the dynamics of T-cell responses, and their interaction with the virus is essential for developing effective therapeutic and preventive strategies against HCV. Furthermore, the comprehension of the role of both structural and NS proteins is crucial for vaccine design, in order to elicit robust and durable immune responses. Moreover, the elucidation of the mechanism by which these proteins interact with the host immune system can assist in the development of novel vaccine strategies that harness the full potential of T-cell-mediated immunity against HCV. Therefore, the objective of this review is to elucidate the significance of these proteins, as well as the various vaccination methods employed, and their impact on current and prospective strategies for developing vaccines against HCV.

## 2. Materials and Methods

The analysis of the literature was conducted in June 2024 using different databases—Pubmed, Science Direct, and Scopus—using the keywords “HCV”, “Hepatitis C”, and “vaccine”. From the papers retrieved (publication date between 2013 and 2024), the title and abstract were screened in order to exclude papers in languages other than English, reviews, and studies without in vivo or in vitro validation. A total of 10,377 papers were identified initially (Pubmed, 1695; Science Direct, 7087; and Scopus, 1595), and the duplicates were excluded. Next, the following inclusion criteria were adopted: (1) open-access papers or available through the Cleveland Clinic library system, (2) DNA- and RNA-based vaccines studies, (3) peptide/protein-based vaccines studies, and (4) computational approached of antigen prediction with in vivo or in vitro validation. After the application of the inclusion criteria, 62 papers (30 protein/peptide-based vaccines studies, 31 DNA-based vaccines, and 1 RNA-based vaccine) were eligible to be included in this review. The data on the cellular response from the selected papers were collected and are described in this review.

## 3. Peptide/Protein-Based Vaccines

### 3.1. Structural Proteins 

Currently, different platforms are available for producing recombinant proteins, offering flexibility and efficiency in biotechnological applications. Examples of these approaches are the prokaryotic (bacteria) or eukaryotic (e.g., yeast, insect, fungal, and mammalian cells) systems, which are well-established methods for recombinant protein expression due to their rapid growth and ease of manipulation [22]. Additionally, for the eukaryotic systems, mammalian cell lines, e.g., Chinese hamster ovary (CHO) cells, offer the advantage of producing complex proteins with appropriate post-translational modifications and are approved for GMP manufacturing [23]. Another commonly used platform uses viral vectors, like adenovirus and lentivirus, modified to carry genes encoding the desired protein for delivery into the host cells, allowing for high-level expression [24]. The virus-like particles (VLPs) represent another innovative strategy, mimicking the structure of viruses in the absence of their genetic material, providing a safe platform for vaccine production and delivery of therapeutic proteins [25]. These diverse platforms cater to the specific needs of different proteins and applications, contributing to advancements in biopharmaceuticals and biomedical research.

In this context, different systems allowed the evaluation of the immune response against potential HCV vaccine target proteins and their role in conferring an effective antiviral response. 

### 3.2. Core Protein

The genotype 1b core protein (expressed using an *E. coli* system) was used for the vaccination of mice in a different study, with two different adjuvants: nonionic copolymer-Pluronic F127 (used for its surfactant and protein stabilizing properties) and the Bacillus of Calmette–Guerin (BCG) [26]. Vaccination with the BCG adjuvant promoted the generation of IFN-γ-secreting cells with a significantly higher frequency, compared to the F127 group, which instead showed a higher frequency of IL4-secreting cells. 

### 3.3. Envelope Glycoprotein 1 and 2

#### 3.3.1. Soluble E2 Expression Using Different Systems and Their Formulation with Various Adjuvants

In a study by Li and colleagues, the authors used an insect cell system for the production of a transmembrane domain-truncated, soluble version of E2 (sE2, genotype 1b) [27]. The mouse vaccination with sE2 induced a high frequency of IFN-γ- and IL-4-secreting cells, regardless of the adjuvant used (i.e., Alum, CpG, or CFA/IFA). However, the induction of both cytokines was higher for the experimental group adjuvanted with Alum/CpG. The same group utilized the same protein construct in a following study, immunizing non-human primates (NHPs) [28]. This approach induced IFN-γ- and IL-4-producing cells in most of the NHPs. Similarly to the mouse study, a significant increase in IFN-γ-secreting cells was observed following the sE2/Alum/CpG and sE2/Alum/MPL vaccinations (with no improvement in IFN-γ secretion upon sE2 stimulation at the time of animal euthanasia at month 8). However, only the sE2/Alum and sE2/Alum/MPL groups significantly demonstrated an IL-4 secretory response sustained over time, suggesting a shift from the Th1 to the Th2 response. Production of IFN-γ and IL-4 following stimulation by intrahepatic lymphocytes in all the vaccination groups at month 8 indicated a successful establishment of a resident T-cell memory.

Czarnota and colleagues designed a chimeric construct containing the E2 region spanning the 412–425 residues (an epitope with broadly neutralizing properties) inserted into the major antigenic loop of the small surface antigen of the hepatitis B virus (sHBsAg) and expressed using a *Leishmania tarentolae* expression system [29]. A more potent cellular response (IFN-γ-secreting cells) in the E2/sHBsAg-immunized mouse group was observed, compared to sHBsAg used alone. 

A significant IFN-γ and IL-4 production was observed when using the insect cell system to express the sE2, mainly when using Alum and CpG as adjuvants—either in mice or NHPs. Furthermore, the use of Alum demonstrated to be effective in eliciting an IL-4-induced sustained response over time in macaques. On the other hand, using the sHBsAg as an adjuvant also improved the IFN-γ production. These studies demonstrated the production of immunogenic HCV proteins by using different platforms, and regardless of the adjuvant used, all of them were able to elicit a specific anti-HCV cellular response.

#### 3.3.2. Antigen Presentation by Viral Vectors

Kord and colleagues used the BacMam baculovirus-based surface display, which both expressed and displayed the E2 gene (of genotype 1a) in mammalian cells and baculoviral envelope [30]. For the mouse vaccination, a prime-boost vaccination regimen was assessed in three groups: a Bac/Bac group (i.e., prime/boost vaccination with the baculovirus vector only), Bac/Pro (prime vaccination with the vector and boost with the recombinant E2 protein), and Bac/DNA (prime vaccination with the vector and boost with the pCDNA plasmid encoding the E2). A final group receiving PBS was included as a negative control. Although all groups produced IFN-γ, the Bac/Bac group showed significantly higher levels. All groups developed similarly low levels of IL-4, indicating a Th1-type polarization of the immune response. 

The simian adenovirus vector, ChAdOx1, and the modified virus Ankara (MVA) were used to express the E2 from genotypes 1 to 6 (Gt1-6) and a modified E2 with or without a deletion of the variable regions (E2Δ123HMW—high molecular-weight) for mouse vaccination in the study by Donnison and colleagues [31]. The monovalent vaccination (Gt1-6 and E2Δ123) elicited a higher E2-specific CD8+ T-cell response (against genotype 1a, but not against genotype 1b), compared to the bivalent scheme (Gt1-6-E2Δ123). Conversely, the bivalent vaccination elicited a greater genotype 3a-specific IFN-γ CD4+ and genotypes 1a- and 1b-specific IFN-γ CD8+ T-cell response, when tested against NS peptide pools (NS3, NS4, and NS5), compared to the monovalent vaccination. When assessing a different vaccination scheme (i.e., prime/boost/boost), the ChAd-Gt1-6-E2Δ123 priming and two boosts with MVA-Gt1-6-E2Δ123 elicited a higher frequency of tumor necrosis factor (TNF)-α- and IFN-γ-producing T cells. 

Zhu and colleagues used replicon particles derived from Sindbis-like virus XJ-160 expressing the E1/E2 (genotype 1b) for the vaccination of mice, where this vaccine construct elicited an E1/E2-specific IL-6, IL-10, and IFN-γ response [32].

All vaccination schemes elicited HCV-specific IFN-γ-producing cells, where the ChAdOx1 and MVA approach demonstrated a broader effect in eliciting both CD4+ and CD8+ T-cell responses across different HCV genotypes. Notably, the use of the Sindbis-like virus resulted in an even broader immune response profile compared to the other approaches. It is noteworthy that, despite combining a vector-based vaccination with recombinant protein- or DNA-based strategies, in the BacMam virus study, the vaccination with the baculovirus vector only (Bac/Bac) provided the highest IFN-γ production.

#### 3.3.3. Antigen Expression by Virus-like Particles

In the study by Earnest-Silveira and colleagues, the authors assessed the elicitation of IFN-γ-secreting T cells by immunization with a vaccine construct composed of VLPs expressing the core, E1, and E2 (genotype 1a) and observed a remarkable core-specific T-cell response in immunized mice [33]. 

The group of Christiansen and colleagues observed, in a mouse vaccination study using VLPs containing the core, E1, and E2 proteins from HCV genotypes 1a, 1b, 2, and 3a as a quadrivalent formulation and different adjuvants (Alum, montanide, or CFA), the induction of the highest number of T cells secreting IFN-γ in the VLP alone-immunized group, followed by the group adjuvanted with Alum, and the lowest in the ones adjuvanted with CFA or montanide [34]. Despite the IFN-γ production by both cell types (CD8+ and CD4+ T cells), this was significantly higher in the CD4+ T cells. Granzyme B was higher in the vaccination with VLP plus Alum, followed by VLP plus montanide. However, the in vitro stimulation of the mouse splenocytes with VLPs for 5 days resulted in an increase in granzyme B production, as assessed by ELISpot, particularly in the VLPs/CFA and VLP/montanide groups. In a following study by the same group, VLPs containing E1 and E2 (genotypes 1a, 1b, 2a, and 3a), intradermally administered to pigs, induced IFN-γ-secreting T cells, with a sustained response over time despite a modest decrease [35]. The assessment of the granzyme B levels showed that this vaccination was able to induce durable cytotoxic T-cell responses in vivo. However, inflammatory cytokine production was not detected (e.g., IL-1a, 1b, IL-6, IFN-γ, and TNF-α), except for IL-12 and a transiently IL-18 induction. 

VLPs can represent a vaccine platform to be considered among the possible future HCV vaccine candidates. In fact, the studies described here demonstrate the VLP-mediated induction of IFN-γ-producing cells. However, differently from what was observed in the previous studies, the use of VLPs and adjuvants did not enhance the immune response in mice. Furthermore, the same vaccination did not induce an inflammatory cytokine production in a larger animal model (i.e., pigs). Considering these results, vaccination using VLPs needs further evaluation, especially for the assessment of their capacity to elicit a robust T-cell response.

### 3.4. Non-Structural Proteins 

The viral structural proteins tend to elicit a more potent humoral response compared to the NS proteins, due to their display on the pathogen’s surface. The immune response to structural proteins is also more closely associated with antibody (Ab)-mediated neutralization and protective immunity, unlike NS proteins, which are the primary target of the T-cell-mediated immune responses. 

#### 3.4.1. P7

Only one study assessed the immunogenicity of p7. Using a nanoparticle-based vaccine containing a panel of six overlapping p7 peptides (of genotypes 1a and 1b) adjuvanted with the cationic liposomal CAF09 for mouse vaccination, the vaccine generated both robust CD4+ and CD8+ T-cell responses, while the vaccination with the p7 protein only induced antigen-specific CD4+ T cells producing IFN-γ, TNF-α, and IL-2 [36]. Furthermore, using a mouse surrogate challenge model, the vaccination with genotype 1a-derived p7 could effectively clear infected liver cells, but not when using a genotype 1b -derived p7 vaccine. This result shows a limitation of this construct in eliciting cross-genotype reactivity. Despite there being few studies focusing on the p7 protein in recent years, this protein represents an interesting target against HCV given its essential role in the assembly and release of infectious HCV viral particles.

#### 3.4.2. NS3 and NS5B

Capone and colleagues used the ChAd3 and MVA vectors to express full-length NS3 and NS5B (genotype 1b) in their vaccine formulation [37]. Besides the HCV NS proteins, the viral vectors also expressed the murine or human invariant chain, a polypeptide that plays a critical role in antigen presentation. This polypeptide has been shown to enhance antigen presentation in MHC class I, leading to an improved and expanded CD8+ T-cell response [38,39]. Vaccination with the ChAd3 significantly improved the magnitude and breadth of T-cell response across genotypes 1b and 3a in immunized mice. Moreover, re-administration of the same vector, and a boost with MVA, efficiently enhanced the CD8+ response. In NHPs immunized with ChAd3 (expressing the human invariant chain), a significantly higher CD8+ T-cell response was observed in comparison with the vector encoding all HCV NS proteins alone and maintained over time. 

#### 3.4.3. NS3, NS4A, and NS4B

Zhu and colleagues observed that, using the adeno-associated virus (AAV) expressing NS3 or NS3/4 (genotype 1b), for mouse vaccination with NS3/4, induced high levels of IFN-γ, IL-2, and IL-4, followed by NS3, with both vaccinations being able to induce effective T-cell responses [40]. In a different approach by Agrawal and colleagues, human adenovirus 5 (Ad5) expressing the NS3 (genotype 1a) (Ad5-NS3), or the Ad5 alone was used, inducing a cellular immunity (characterized by proliferation and cytokine secretion) in vaccinated mice against core, NS3, and NS5 [41]. However, IFN-γ production appeared to be higher against the NS3 antigen in Ad5-NS3-immunized mice, and IL-10 responses against the core, NS3, and NS5 were significantly increased in the same group.

#### 3.4.4. NS2, NS3, NS4A, NS4B, NS5A, and NS5B

Using the Semliki Forest Virus (SFV) as a vector expressing all the NS proteins (genotype 1a) or parts of them (i.e., NS3/NS4A and NS5A/B), Ip and colleagues observed that a potent NS3-specific CD8+ T-cell response was elicited, mainly in mice immunized with SFV encoding NS3/4A (increased effector memory and effector T cells), compared to those encoding NS2-NS5B (associated with a central memory T-cell response) [42]. A different study also used SFV, expressing the NS3/4A or NS5A/B (genotype 1a) with or without the antigen sigHELP-KDEL (a T cell and endoplasmic reticulum targeting signal) [43]. It was found that the vaccinations of mice induced equal levels of HCV-specific CD8+ T cells with functional effector and memory phenotypes, suggesting that the inclusion of sigHELP-KDEL in the candidate HCV vaccines does not further augment HCV-specific T-cell responses. 

It was possible to observe an augmented T-cell immune response in all the studies described above, showing that using viral vectors could be an effective strategy for vaccine development. Noteworthy, most of these works used NS3 and NS4, the HCV proteases responsible for the cleavage at four sites of the HCV polyprotein, essential for viral replication (Figure 1), highlighting the relevance of these proteins as vaccines and drug targets.

### 3.5. Combination of the Structural and Non-Structural Proteins

Viral Vector Protein Expression.

#### 3.5.1. E1, E2, NS3, and NS4A

Among the studies that used different vectors to immunize NHPs, the group of Wen and colleagues observed that there was no detectable T-cell response after priming with E1/E2/NS3-expressing HCVpp of genotype 1a [44]. In contrast, priming with the replication-defective vaccinia virus (VV) (Tiantan strain, rNTV) was able to elicit a T-cell response (with a decline at week 32 but enhanced following a boost with the recombinant adenovirus (rAd) at week 36). The production of Th1 cytokines (IL-2, IL-12, TNF-α, and IFN-γ) was higher in the rNTV/rAd vaccination group. The rNTV/rAd group also showed enhancement of Th2 cytokines, correlating with the strongest HCV-specific humoral response, although the rNTV/rAd group had a higher IL-6 level. The use of the E1/E2/NS3-expressing HCVpp did not improve cellular immunity (as also observed when boosting with the same strategy). However, vaccination with the viral vector elicited the highest cellular response, especially when priming with the rNTV (which responded to all core, E1, E2, and NS3 peptide pools). 

A construct with a similar protein composition named mosaic proteins was designed by Yusim and colleagues [45]. The mosaic proteins are in silico-designed antigens that resemble natural proteins based on the E1, E2, NS3, and NS4A (genotypes 1a and 1b) proteins and encoded by adenovirus serotype 35 (Ad35). The vaccine generated in mice a T-cell response that was higher in the case of the vaccination with the mosaic proteins, compared to wild-type proteins (full-length from genotypes 1a and 1b). One possible explanation for this could be that the mosaic proteins displayed more distinct 9-mer potential epitopes as opposed to the wild-type proteins, or the delivery of particular epitopes in the BALB/c MHC (H-2^d^ haplotype) was more effective in this mouse strain compared to other mouse wild-type strains.

#### 3.5.2. E1, E2, NS3, NS4A, NS4B, NS5A, and NS5B

The E1, E2, and NS3-NS5B proteins (genotypes 1b and 6a) were used in the experimental vaccine described by Luo and colleagues for the vaccination of mice [46]. Two adenovirus vectors were used, Sad23L and Ad49L (previously used to develop immunogens for the Zika virus and the severe acute respiratory syndrome coronavirus 2), with low serologic prevalence compared to other adenoviruses such as Ad5 [47,48]. Eight vaccine constructs were designed: HCV-structural protein (E)-based (Sad23L-E-1b, Sad23L-E-6a, Ad49L-E-1b, and Ad49L-E-6a) and HCV-NS protein-based (Sad23L-NS-1b, Sad23L-NS-6a, Ad49L-NS-1b, and Ad49L-NS-6a). All vaccine constructs elicited broad and highly specific IFN-γ secreting T-cell responses in a dose-dependent manner, compared to the control groups (PBS inoculation or the viral vector with no HCV protein expression). Furthermore, the prime-boost vaccinations using Sad23L/Ad49L expressing the same antigens elicited higher HCV-specific T-cell responses compared to a single-shot HCV vaccine, as well as a higher and broader cross-reactive cellular immunity. The HCV vaccines elicited a robust NS3–1b-specific CD8+ T-cell response in the spleen and liver, with a high proportion of tissue-resident memory and effector memory cells in the liver. These cells can produce higher levels of antiviral cytokines, playing a crucial role in rapid protection against hepatotropic HCV infection.

#### 3.5.3. Full-Length HCV Genome

The full-length HCV genome sequence (genotype 1a), expressed by the MVA vector in combination with the C6L gene, associated with IFN-β expression, or a C6L-depleted version (MVA-ΔC6L), were administered to mice in the study of Marín and colleagues [49]. Vaccination with MVA-ΔC6L or MVA-C6L elicited a high level of HCV-specific CD8+ T-cell-mediated immune response (when considering the total number of cells producing IFN-γ, TNF-α, and/or IL-2, and expressing the CD107a degranulation marker). The response was largely mediated by HCV-specific CD8+ T cells with lower levels of HCV-specific CD4+ T cells. The MVA-C6L and MVA-ΔC6L HCV-specific CD8+ T-cell adaptive immune response was directed preferentially against p7 and NS2 (higher for MVA-C6L) and, to a lesser extent, toward NS3 (higher for MVA-ΔC6L). HCV-specific CD8+ T cells producing CD107a/IFN-γ/TNF-α, IFN-γ/TNF-α/IL-2, or IFN-γ/TNF-α were the most abundant cell populations, similarly induced by MVA-C6L and MVA-ΔC6L. The two versions of the vaccine elicited similar levels of T-cell response, suggesting a negligible role of C6L in contributing to the T-cell response. It is noteworthy that the vaccine formulation efficiently elicited a cytotoxic response. Furthermore, the memory CD8+ T-cell response was elicited at a similar magnitude in both vaccination groups. 

The study of Delft and colleagues used the ChAdOx vector expressing conserved antigens spanning the entire HCV genome (of genotypes 1–6) for mouse vaccination [50]. The vaccine elicited strong cross-reactive HCV-specific CD4+ and CD8+ T-cell responses, producing IFN-γ and TNF-α. In silico epitope analysis revealed that the conserved antigens contain multiple epitopes described in natural HCV infection, which can assist in predicting immunogenicity in humans.

### 3.6. Other Approaches

#### 3.6.1. Core, E1, E2, and NS3

Two different studies by Martínez-Donato and colleagues assessed the immune response to the core/E1/E2/NS3 (genotype 1b). In the first work, the construct MixprotHC (composed of proteins adjuvanted with Alum) induced a potent CD4+ and CD8+ response in mice against all antigens and an IFN-γ response in monkeys [51]. Noteworthy, the MixprotHC showed a remarkable reduction in viral titer in mice challenged with VV expressing the HCV structural proteins. Moreover, monkeys vaccinated with Alum only did not show any response. In the second work using polyhydroxybutyrate beads displaying the core protein on their surface formulated with Alum (Beads–Core) mixed with the E1, E2, and NS3 (genotype 1b), the Beads–Core mouse vaccination elicited a CD4+ response, with 100% of the cytokine-secreting cells being positive for IFN-γ secretion after three doses [52]. Following challenge with the VV, the use of the Beads–Core vaccine was associated with a remarkable reduction in viral titer (for either the VV expressing the core only or the other HCV structural proteins) compared to the other groups. Both studies presented vaccine formulations capable of inducing a functional cellular immune response, able to control viremia after challenge. While the second work used a murine model for the vaccination studies, the first also used the NHP model, which possesses a more similar immune system to humans. Nonetheless, both vaccine formulations showed promising results to be considered in future clinical trials.

A vaccine designed by Liu and colleagues using the **core/E1**/E2/NS3 proteins (genotype 1b), with multihydroxylated fullerene (C_60_(OH)_22_) as an adjuvant for mice vaccination, promoted an increased anti-HCV/E2 IFN-γ-response [53]. Noteworthy, the use of this adjuvant could help in the antigen spare (adjuvant optimal dose range of 0.1–2 μg/mouse for 5 μg of HCV-E2 antigen/mouse).

#### 3.6.2. E1, E2, NS4B, NS5A, and NS5B

When using 6 peptides derived from conserved regions of the E1_315–326_, E2_412–423_, NS4B_1771–1790_, NS5A_2121–2140_, and NS5B_2941–2960_ from genotypes 2a and 4a for the vaccination of mice, Dawood and colleagues observed that the IFN-γ-secreting CD4+ and CD8+ T cells appeared earlier when using the 1.6 and 16 μg dose vaccinations (within 2 weeks, and it remained stable over time) [54]. On the other hand, in the 800 ng dosage group, this response appeared only after 6 weeks, and it declining over time. Noteworthy, the cellular response was maintained for over 20 weeks, indicating that this vaccination could offer long-lasting protection. 

## 4. DNA-Based Vaccines

DNA and RNA vaccines represent innovative approaches to vaccination that have gained significant attention in recent years due to their potential, offering several advantages over traditional vaccine platforms. Antigens delivered through DNA vaccines are transcribed and directly translated by the host and undergo native post-translational modifications. This antigen is then presented to the immune system. One of the pioneering studies in DNA vaccination demonstrated protection against the influenza virus in mice using a DNA plasmid encoding the viral hemagglutinin protein [55]. Differently, RNA vaccines use messenger RNA (mRNA) that can be directly translated to the desired antigen. Similar to DNA vaccines, mRNA vaccines are taken up by host cells and translated into antigens, processed, and are presented on the cell surface, leading to the activation of the immune response. Notably, mRNA vaccines have gained widespread attention during the COVID-19 pandemic, with the development of SARS-CoV-2 Spike encoding vaccines by Pfizer-BioNTech and Moderna [56].

One of the key advantages of DNA and RNA vaccines relies in their ability to induce both humoral and cellular immune responses, leading to a robust and comprehensive immune response [57]. This versatility is particularly important for combating intracellular pathogens and viruses, where cellular immunity plays a crucial role. Moreover, DNA and RNA vaccines offer several logistical advantages over traditional vaccines. They are relatively easy and cost-effective to design and produce, making them particularly suitable for a rapid response to emerging infectious diseases. Additionally, these vaccines do not require live pathogens, reducing the safety concerns associated with traditional vaccine platforms. Furthermore, while some of these vaccines are formulated with adjuvants, others are capable of eliciting a potent immune response as unadjuvanted formulations. 

The immune response elicited by DNA and RNA vaccines resembles the natural infection. Antigen-presenting cells, such as dendritic cells, uptake the genetic material and present the encoded antigens to T cells, which are activated and in turn help in stimulating B cells and Ab production. Memory T and B cells are also generated, providing long-term immunity against future encounters with the pathogen [58]. Both vaccines represent promising tools for preventing infectious diseases and have garnered considerable interest for their potential to address current and future public health challenges. 

### 4.1. Structural Proteins

#### 4.1.1. Core

In the study of Hartoonian and colleagues, the authors observed that the co-inoculation of pMIP-3beta (DNA plasmid encoding the MIP-3beta gene, implicated in the modulation of the dendritic cell activity) and pCore (DNA plasmid for the core of genotype 1) in mice enhanced the production of antigen/epitope-specific IFN-γ 1.5-fold higher than with pCore alone [59]. However, the IL-4 release did not significantly differ between the control and the test groups, demonstrating an amplified Th1 polarized response. ELISpot assay showed a drastic difference in the number of IFN-γ-secreting cells in favor of mice co-vaccinated with pMIP-3beta compared to pCore alone, indicating the role of MIP-3beta chemokine as an adjuvant. The results of lymphoproliferative responses revealed a significant antigen-specific increase in the number of splenocytes of mice co-vaccinated with pMIP-3beta and pCore. pMIP-3beta coadministration induced a significantly higher percentage of antigen/epitope specific granzyme B release compared to the pCore group. 

Using a plasmid encoding the core and HBsAg (pCHCORE) in a mouse vaccination study, Yazdanian and colleagues observed a high lymphocyte proliferation and cytotoxic T-lymphocyte (CTL) activity, similar to that elicited by vaccination with the core protein [60]. However, DNA vaccination elicited significantly higher levels of IFN-γ than the protein-based vaccination, while the protein-vaccinated group showed an elevated production of IL-4, compared to the DNA-immunized group.

The Lambda-ZAP^®^-CMV vector (Stratagene, La Jolla, CA, USA)—which enables high-levels of eukaryotic expression and gene delivery—expressing the core protein was used by Saeedi and colleagues in addition to DNA-based vaccination (of genotype 1) for mouse vaccination [61]. The lymphocyte proliferation was higher in the heterologous vaccination with DNA/Lambda than homologous (Lambda/Lambda and DNA/DNA) and heterologous (Lambda/DNA) prime-boost groups. Also, for the DNA/Lambda and Lambda/Lambda groups, the cytotoxic T lymphocyte and IFN-γ production was similar. However, the IL-4 production was higher in the Lambda/Lambda group, compared to the other ones. 

The use of the core protein, a multifunctional protein but mainly related to forming the viral capsid to surround and protect the genomic viral RNA, was assessed in the above-described works. All the DNA vaccine formulations were able to elicit a cellular immunity. These vaccines, adjuvanted or combined with viral vectors, provided better results. In the case of DNA-based vaccines, using viral vectors such as the Lambda-ZAP^®^-CMV might provide a more efficient DNA delivery, consequently enhancing the T-cell response, especially for heterologous vaccinations. Noteworthy, in the work of Saeedi and colleagues, the DNA vaccination elicited significantly higher levels of IFN-γ, compared to the protein-vaccinated group (even though the protein-based vaccination elicited elevated levels of IL-4). A limitation of these studies is their main exclusive focus on the IFN-γ and IL-4 cytokine profiles.

#### 4.1.2. Envelope Glycoprotein 1 and 2

A plasmid encoding E1, E2 (of genotype 1b), and IMX313P, a complement inhibitor, was used by Masavuli and colleagues for a mouse vaccination study [62]. The vaccination with the E1, E2, or E1/E2 ectodomain elicited a low IFN-γ response, as determined by ELISpot, while DNA E1/E2/IMX313P-based vaccination induced a higher IFN-γ response. Here, the IFN-γ levels after DNA vaccination were higher compared to the protein-based vaccination. This outcome is similar to the previously described core protein vaccination approach. These results suggest that DNA vaccination can induce a higher IFN-γ response compared to protein-based immunization.

In their study, Nadeem and colleagues transfected peripheral blood mononuclear cells (PBMCs) from HCV-uninfected human donors with DNA encoding the E1 and E2 proteins (genotype 4a), then tested this construct in vivo to assess its ability to stimulate cellular responses [63]. After successfully transfecting and expressing E1 and E2 on PBMCs, mice were vaccinated with the DNA construct, resulting in a significant early increase in CD4+ T lymphocytes compared to the control group (empty plasmid), while CD8+ T cells showed no significant change. Although the HCV-specificity of the primed T-lymphocyte response needs further assessment, these findings are promising for future studies on both murine models and human PBMCs from antigen-naïve and previously infected patients. 

### 4.2. Non-Structural Proteins

#### 4.2.1. NS2 

In the study of Gorzin and colleagues, the inclusion of the full-length NS2 (of genotype 2a) in their vaccine formulation elicited high NS2-specific lymphocyte proliferation when co-administering IL-12 in mice, even with higher levels of IFN-γ secretion, but not of IL-4, with a booster [64]. The full-length NS2 protein from genotype 2a combined with IL-12 as an adjuvant demonstrated enhanced lymphocyte proliferation and a selective increase in IFN-γ secretion, without affecting IL-4 levels. This result suggests that the IL-12 skews the immune response towards a Th1 type. The selective promotion of IFN-γ over IL-4 secretion also indicates a strategic benefit in possibly reducing the risk of Th2-mediated enhancement of viral infections, which can complicate vaccine efficacy. 

#### 4.2.2. NS3 

In the study of Ratnoglik and colleagues, a series of DNA vaccines expressing the NS3 (of genotype 1b) were designed and tested in a mouse immunization study [65]. To inactivate the enzymatic activity of the NS3 protease, single-point mutations into each of the catalytic triad of NS3 (H57A, D81A, and S139A) and the NTPase/RNA helicase domain (K210N, F444A, R461Q, and W501A) were introduced. A strong induction of IFN-γ production was achieved after introducing the single-point mutations, suggesting that the presence of these mutations may contribute to the exposure or boost the response against some epitopes and thus elicit a robust immune response. 

Moreover, in the study of Pouriayevali and colleagues, double-mutants lacking both serine protease and NTPase/RNA helicase activities, NS3 (S139A/K210N), NS3 (S139A/F444A), NS3 (S139A/R461Q), and NS3 (S139A/W501A), induced a higher IFN-γ production and a stronger CTL activity in vaccinated mice. Using a different construct based on the NS3 (of genotype 1a) elicited a significantly higher lymphocyte proliferation, with induction of both specific IFN-γ- and IL4-producing cells [66]. In a different study, using *Artemisia annua* polysaccharides as adjuvant in the Bao and colleagues NS3-vaccine formulation, the level of serum IFN-γ secretion in vaccinated mice was significantly higher than that of IL-4 secretion at a dosage of 0.5 mg/mL of the adjuvant (compared to 1 and 0.1 mg/mL) [67]. 

Using a new strategy, Chi and colleagues used the bacterial ghost (BG), a novel DNA vaccine delivery system in their vaccine formulation (BG-NS3-Ii) based on NS3 and the murine invariant chain (explained elsewhere in this review) for mouse vaccination [68]. The BGs maintained the cellular morphology and antigenic structures of native bacteria. BGs with recombinant DNA leverage the superior bioavailability of DNA-based vaccines and ensure high expression rates of DNA-encoded antigens in antigen-presenting cells (APCs), such as macrophages and dendritic cells. Furthermore, BGs, similar to living bacteria, function as natural adjuvants, aiding in the recruitment of innate immunity regulators and activation of APCs [69]. To compare the capacity of BG-NS3-Ii and the plasmid expressing only the NS3-Ii to induce NS3-specific T-cell responses, a cytotoxicity assay was performed 2 weeks after the last vaccination using the splenocytes from each mouse-vaccinated group. The mice immunized with BG-NS3-Ii exhibited higher cytotoxicity levels (25%) compared to those vaccinated with the NS3-Ii plasmid (17%), indicating that BG-NS3-Ii vaccination induces a stronger CTL response specific to NS3-Ii. 

The use of *Artemisia annua* polysaccharides demonstrated that the adjuvant dosage plays a critical role in the balance and type of elicited immune response; a medium dose (0.5 mg/mL) was the most effective in boosting IFN-γ over IL-4, indicating a shift towards a Th1-type immune response, which is desirable for combating viral infections like HCV. The BGs significantly enhance the immune response, inducing higher NS3-specific CTL responses in mice (BG-NS3-Ii compared to traditional plasmid-based vaccines). BGs act as natural adjuvants, improving antigen presentation and immune activation while being safer than live bacteria, suggesting a promising and safe vaccine strategy for HCV and other infectious diseases. These results underscore the importance of the physical form of the adjuvant in vaccine formulations, potentially influencing antigen presentation and subsequent T-cell activation.

#### 4.2.3. NS3 and NS4A

In the study of Levander and colleagues, the authors showed that the use of a different adjuvant, the avian hepatitis B core antigen (HBcAg), for mouse vaccination with the DNA encoding the NS3/4A (of genotype 1a) significantly improved the induction of NS3-specific CD8+ T cells [70]. Furthermore, repeated vaccinations boosted the breadth and magnitude of the NS3-specific T-cell response. The co-expression of the murine IL-12 with the NS3/4A also improved the immunogenicity. Noteworthy, using fragmented HBcAg was more efficient in enhancing the T-cell response than using the full-length HBcAg. 

#### 4.2.4. NS5A

Holmström and colleagues showed that mouse vaccination with two NS5A peptides binding to H-2Db (9-mer) and H-2Kb (8-mer) epitopes elicited high levels of IFN-γ secreted by NS5A-specific CD8+ T cells and IL-2 production [71]. In the study of Wijesundara and colleagues, the NS5B from HCV genotypes 1b and 3a was formulated as a cocktail for mouse vaccination and was able to elicit high responses to different NS5B peptides as tested by ELISpot, and larger numbers of NS5B-specific IFN-γ-secreting T cells [72]. An in vivo evaluation suggested that the cocktail vaccination increased the breadth and magnitude of CTL and Th cell responses more effectively than global consensus vaccination regimens (cytolytic DNA encoding genotype-specific antigens).

The NS5A-based vaccine formulation used by Holmström and colleagues was particularly effective in inducing high levels of IFN-γ and IL-2 production by CD8+ T cells. This targeted approach harnesses the power of epitope-specific vaccination, which can be crucial in eliciting a strong cytotoxic T-cell response. This method success underscores the importance of epitope selection in vaccine design, potentially offering a blueprint for developing highly targeted vaccines that optimize T-cell-mediated immunity. The cocktail vaccine containing peptides from NS5B resulted in high responses and significant numbers of NS5B-specific IFN-γ-secreting T cells. This approach is in contrast with the more conventional strategy of using global consensus sequences by demonstrating that a mixture of genotype-specific antigens can enhance the breadth and magnitude of both CTL and helper T-cell responses. The in vivo effectiveness of this cocktail vaccine suggests that incorporating multiple variants of the same protein can be more effective than a single consensus sequence, enhancing the vaccine capacity to overcome viral diversity.

#### 4.2.5. NS3, NS4A, NS4B, NS5A, and NS5B

In the study of Gummow and colleagues, the PRF and the vesicular stomatitis virus protein G (VSVG) with the NS3, NS3/4A, NS4B/5B, NS3/NS4B/NS5B, or NS3/NS4/NS5B (of genotype 3a) were used to create different constructs to be used in a mouse vaccination study [73]. A higher NS3-specific cell-mediated immune response following vaccination with NS3, compared to NS3/4A, was observed, which was enhanced when also using PRF and VSVG. Moreover, the NS3/PRF vaccination increased the frequency of single-, double-, and triple-cytokine-producing CD8+ T cells. The NS3/4/5B polyprotein generated higher HCV-specific IFN-γ responses against the individual antigens, compared to the plasmid encoding either NS3 or NS4B/5B. 

Using a construct based on the NS3/4A, NS4B, NS5A, and murine IL-28b as an adjuvant, Lee and colleagues observed a potent T-cell immune response, with a plateau when using a 20 μg/dose of the vaccine (other dosages were 5 and 40 μg) [74]. This may be related to a saturation of the immune response in humans. This dosage induced a lower NS3/4A-specific T-cell response, compared to vaccination with NS3/4A or administration of IL-28 alone. Using murine IL-28b as an adjuvant, an induction of the T-cell immune responses twice as great when compared to the vaccination without the adjuvant was observed.

The administration of mesenchymal stem cells (MSC) alone or followed by vaccination in mice with NS3/NS4A/NS4B/NS5A/NS5B (genotype 1b) by Masalova and colleagues elicited higher T-cell proliferation and IFN-γ production [75]. Furthermore, using specific stimulators (NS3, NS4B, and NS5A peptides), the cellular immune response (proliferation and IFN-γ production) was observed only in the group immunized with the DNA-based vaccine or in the one co-administered with MSC. When assessing the IFN-γ levels in the serum of immunized mice, vaccination with the HCV-derived DNA led to an increase in the serum level of IFN-γ, whereas a reduction was observed with the administration of MSC prior to the DNA-based vaccine administration.

The studies described here present various immune modulation approaches, including the use of additional immune-stimulatory proteins, PRF and VSVG, the incorporation of cytokine adjuvants, and the strategic administration of cellular therapies like MSC. Each of these methods shows unique outcomes in terms of eliciting cell-mediated immune responses and cytokine production, particularly IFN-γ, which is crucial for effective viral clearance. The study involving different NS3-based constructs (e.g., NS3 alone, NS3/4A, NS3/4/5B) reveals that the complexity of the construct can affect the immune response. Vaccination with NS3 alone, especially when adjuvanted with PRF and VSVG, generated a higher cell-mediated response than more complex constructs like NS3/4A. This suggests that while larger polyproteins might cover more viral antigens, smaller constructs might be more effective in inducing specific T-cell responses. Notably, the NS3/4/5B elicited higher specific IFN-γ responses against individual antigens than either NS3 or NS4B/5B alone, indicating that certain combinations might optimize the immune response to multiple targets. On the other hand, the addition of murine IL-28b as an adjuvant with an NS3-based construct showed a dosage-dependent saturation of the immune response. Higher or lower dosages either did not enhance or actively reduced the specific T-cell response, highlighting the necessity of fine-tuning vaccine dosages for optimal efficacy. This saturation effect suggests a potential limitation in the adjuvant’s ability to boost the immune response beyond a certain threshold. The administration of MSCs showcased an intriguing interaction with the immune response to DNA-based vaccination. While MSCs alone or in combination with DNA vaccines enhanced T-cell proliferation and IFN-γ production, pre-administration of MSCs before the DNA vaccine surprisingly reduced the levels of IFN-γ in the serum. This result may reflect the immunomodulatory properties of MSCs, which can sometimes exert suppressive effects on immune activation depending on the context of administration. This finding underscores the complexity of using cellular therapies in vaccination strategies and the need for careful scheduling and dosing of such treatments.

### 4.3. Viral Vectors in DNA-Based Vaccines

#### NS3, NS4A, NS4B, and NS5B

In the study by Tan and colleagues, the authors used three common viral vectors—Ad5, MVA, and VV—expressing the NS3/4A (of genotype 1a) for a mouse vaccination study to compare the magnitude and phenotypic characteristics of CD8+ T cells [76]. The vaccination elicited a broad CD8+ T-cell response—after priming with DNA encoding NS3/4A. The Ad5 vaccination was significantly more immunogenic with, however, a lower magnitude of memory phenotype, although these cells were less multifunctional than those induced by VV and MVA vaccinations. Furthermore, after challenge with a murine herpes virus expressing NS3, no virus was detected in the spleen, suggesting an effective viral control. Following the vaccination of mice with ChronVac-C (containing the full-length codon optimized NS3/4A of genotype 1a) and MVATG16643 (MVA vector encoding the genotypes 1b NS3, NS4A/B, and NS5B) by Fournillier and colleagues, both constructs equally induced IFN-γ-producing T cells, with the response being cross-reactive between genotypes 1a and 1b [77]. However, the IFN-γ and IL-2 responses against the NS5B were not detected in all the vaccinated groups. Using a surrogate challenge model based on recombinant *Listeria* monocytogenes expressing the NS3 protein, it was observed that the prime/boost schedules (ChronVac-C/MVATG16643 and MVATG16643/MVATG16643) were the most efficient at reducing *Listeria* titers. Overall, the heterologous ChronVac-C prime and MVATG16643 boost resulted in higher frequencies of HCV-specific IFN-γ- and IL-2-producing T cells than that obtained with the separate vaccines.

In the study of Mekonnen and colleagues, vaccination with AAV encoding the NS5B or the DNA encoding NS5B (of genotype 3a) and perforin (PRF) elicited a higher number of NS5B-specific CD8+ T cells in the liver and spleen of vaccinated mice, being durable after 42 days from the final vaccination in the absence of priming or boost with NS5B/PRF [78]. Further analyses showed that the NS5B-specific CD8+ T cells were tissue-resident memory cells in the liver and effector memory T cells in the spleen. Finally, the assessment of the T helper responses against a pool of NS5B peptides revealed that the vaccination containing AAV/NS5B elicited the highest response.

The results of these studies provide a significant understanding of the immunogenic potential and efficacy of various viral vector-based vaccines combined with DNA-based vaccines. Each vector—Ad5, MVA, VV, and AAV—demonstrated unique immunological responses and efficacy profiles that contribute to a broader insight into vaccine design and strategies against HCV. The ability of the ChronVac-C and MVATG16643 vaccines to induce cross-genotype T-cell responses is particularly crucial for HCV, given the virus’s genetic diversity. Noteworthy, in the work by Mekonnen and colleagues, the durable nature of the immune response, especially the persistence of NS5B-specific CD8+ T cells for over 42 days post vaccination, is significant. Long-lasting immunity is critical for a successful vaccine, particularly for chronic infections.

### 4.4. Vaccination of Non-Human Primates

#### NS3, NS4A, NS4B, NS5A, and NS5B

In the study of Latimer and colleagues, the vaccination of rhesus macaques with either NS3/4A, NS4B, NS5A, or NS5B (of genotypes 1a and 1b) showed that antigen-specific IFN-γ response against these peptides was variable, ranging from undetectable to 1783 spot-forming units (SFU)/10^6^ PBMCs, 3 to 726 SFU, 100 to 796 SFU, and 66 to 2000 SFU, respectively [79]. This vaccination gave rise to IFN-γ-secreting HCV-specific CD8+ T cells, with HCV-specific CD4+ Th1 cells also being detectable. The production of TNF-α, IL-2 was also induced. Furthermore, an increased granzyme B production within the CD8+ T cells and significant CTL activity was observed by 2 weeks post vaccination. This DNA vaccine demonstrated strong immunogenicity, broad T-cell reactivity, and phenotypes associated with protection against chronic HCV infection.

In a study by Callendret and colleagues, two chimpanzees with chronic HCV infection and under an NS5B inhibitor treatment were primed with Ad6-NSmut (expressing the NS3 to inactivated NS5B polymerase of genotype 1b), then boosted with VV Ankara expressing the same proteins [80]. One animal was boosted with DNA expressing the NSmut. CD8+ T cells with antiviral effector functions were induced but could not contain the break-through replication of a virus with resistance to the NS5B DAA. Also, most of the CD8+ T cells expanded by vaccination did not recognize the circulating persistent virus. Approximately half of these cells detected systemically targeted vaccine-specific epitopes, and all other peripheral HCV-specific expanded CD8+ T cells were present in the liver before vaccination. 

One limitation of both studies was the use of only two animals. However, the use of NHPs is restricted due to ethical, practical, and cost-related concerns. Furthermore, the emergence of drug-resistant HCV variants in both animals was unexpected in the work of Callendret and colleagues. However, this vaccine suppressed HCV replication during the acute phase of infection and may be able to prevent the occurrence of a persistent infection, which needs further investigations.

### 4.5. Combination of the Structural and Non-Structural Proteins

#### 4.5.1. Core, E2, NS3, and NS5B 

In the study of Pishraft-Sabet and colleagues, a construct (a multi-epitope DNA- and peptide-based vaccine) containing specific CD8+ T-cell epitopes (HLA-A2 and H2-Dd) from core_132–142_, NS3_1073–1081_, and NS5B_2727–2735_, a Th CD4+ epitope from NS3_1248–1262_ and a B-cell epitope from E2_412–426_ (genotype 1a and 1b) was used for mouse vaccination [81]. Following vaccination, IFN-γ-producing T cells increased over time but mainly in the group immunized with the homologous peptide plus adjuvant regimen, compared to the DNA-based regimen. The levels of IL-4 were very low for both vaccination regimens; however, the homologous peptide plus adjuvant regimen produced significantly higher levels of IL-4, compared to the DNA/peptide regimen. In a subsequent study from the same group, using a polytope plasmid (multi-epitope encoding DNA), codifying for the same proteins in fusion with the N-terminal fragment of the heat shock protein gp96 (NT-gp96) in a mouse vaccination study, induced significant E2 epitope-specific IFN-γ-, TNF-α-, and IL-2-CD8+ T-cell responses, compared to the group immunized with the plasmid alone [82]. 

The same construct was used in a different study conducted by Ansari and colleagues using the CpG as an adjuvant in a homologous mouse vaccination study with *Leishmania tarentolae* expressing the core/E2/NS3/NS5B [83]. This approach showed significantly higher levels of IFN-γ production, compared to the same construct without the adjuvant. Moreover, heterologous DNA/*L. tarentolae* vaccination showed the highest core-, E2-, NS3-, and NS5B-specific IFN-γ secretion compared to the *L. tarentolae* homologous vaccination. The IL4-production was higher in the homologous *L. tarentolae* vaccination group, compared to the heterologous DNA/*L. tarentolae group*, but not for IL-17 and TNF-α production.

The use of a homologous peptide plus adjuvant regimen produced not only a sustained increase in IFN-γ-producing T cells over time but also yielded higher levels of IL-4 compared to a DNA-based approach, suggesting a more balanced Th1/Th2 response, which can be crucial for an effective viral clearance. Incorporating the NT-gp96 into a polytope plasmid further enhanced the responses, inducing significant levels of IFN-γ, TNF-α, and IL-2 from CD8+ T cells, underscoring the potential of molecular adjuvants inclusion in the vaccine regimens. Similarly, using CpG in a homologous vaccination regimen with *L. tarentolae* expressing HCV proteins, an increase of the IFN-γ level was observed, while a heterologous DNA/*L. tarentolae* strategy elicited the highest specific IFN-γ secretion across core, E2, NS3, and NS5B. Interestingly, while the homologous *L. tarentolae* vaccination increased IL-4 production, suggesting a propensity towards a Th2-type response, it did not affect IL-17 and TNF-α levels, highlighting the complex interplay between vaccine constructs and immune modulation. These findings underline the importance of strategic adjuvant inclusion and vaccination regimen design to optimize the elicitation of desired immune profiles for effective vaccine-induced protection against HCV.

#### 4.5.2. Core, E1, E2, p7, NS2, and NS3

Using the alphavirus-based DNA-launched replicon (DREPs) vectors expressing the core/E1/E2/p7/NS2/NS3 (of genotype 1a) and the MVA encoding a nearly full-length HCV genome (7.9-kbp DNA fragment of the HCV ORF from genotype 1a), Marín and colleagues observed that the DREP-prime/MVA-boost vaccination of mice induced a potent HCV-specific CD4+ and CD8+ T-cell adaptive and memory immune response that was significantly higher compared to the group receiving the homologous MVA vaccination [84]. However, the homologous vaccination elicited HCV-specific CD8+ T cells that cross-reacted with pooled peptides from HCV genotype 1b. The DREP-e-HCV/MVA was the most immunogenic combination, eliciting the highest levels of CD4+ and CD8+ HCV-specific immune responses, and also inducing HCV-specific memory T cells with an effector memory phenotype (CD127+ CD62L−). With DREP prime followed by an MVA boost regimen, the production of a potent HCV-specific adaptive and memory immune response was observed and significantly outperforming the homologous MVA-based immunization. The heterologous DREP/MVA strategy was particularly effective, inducing robust CD4+ and CD8+ T-cell responses and generating effector memory T cells, indicating a durable immune memory that can be beneficial for long-term protection against HCV.

#### 4.5.3. Core, E1, E2, NS2, NS3, NS4, and NS5

In the study of Wada and colleagues, two different plasmids, CN2—encoding the core/E1/E2/NS2 (of genotype 1b)—and N25—encoding the E2/NS2/NS3/NS4/NS5 (of genotype 1b)—were used for the vaccination of mice [85]. The vaccination elicited an IFN-γ production and cytotoxicity in both the regimens, where the CD8+ T-cell response was much stronger than the CD4+ T-cell response. Using mice expressing HCV cDNA (i.e., expressing the HCV core, E1, E2, and NS2 proteins consistently for at least 600 days and developing chronic active hepatitis, steatosis, lipid deposition, and hepatocellular carcinoma), vaccination with the N25 resulted in a reduced expression of the HCV proteins in the liver and a decreased expression of the HCV core protein due to the CD8+ and CD4+ T-cell responses. Cellular immune responses to structural and NS proteins were reduced in N25 DNA vaccine-immunized mice expressing the HCV cDNA, while CD4+ and CD8+ T-cell responses to HCV antigens were abolished following CN2 DNA vaccination. The use of two plasmids, CN2 and N25, showed that both regimens effectively generated IFN-γ production and cytotoxicity, with a stronger response derived from CD8+ T cells and not from CD4+ T cells. However, in a mouse chronic HCV model, the N25 vaccine notably reduced the HCV proteins expression in the liver with a moderate immune response, suggesting a potential therapeutic benefit. In contrast, the CN2 vaccine did not elicit significant T-cell responses in this model.

### 4.6. HCV DNA Delivered by Viral Vectors

#### Core, E1, E2, and NS3

Different vectors—MVA, SFV, and Ad5—encoding the core/E1/E2/NS3 (of genotype 1b) and DNA-based vaccines were used for the vaccination of rhesus macaques by Rollier and colleagues, inducing core-, E1-, E2-, and NS3-specific T-cell responses after the Ad5 or MVA injections, including higher IFN-γ production than IL-4, while SFV and MVA induced stronger E2-specific T-cell responses in either a DNA-based and/or vector prime-boost vaccination regimens [86]. 

### 4.7. Full-Length HCV Genome

Another study using MVA encoding the nearly full-length HCV genome (7.9-kbp DNA fragment of the HCV ORF from genotype 1a) showed that homologous DNA/vector or heterologous prime-boost approaches in mice were able to stimulate a broad HCV-specific CD4+ and CD8+ T-cell response, mainly mediated by CD8+ T cells [87]. However, the CD4+ T cells were highly polyfunctional and were crucial for the maintenance and expansion of antigen-specific CD8+ T-cell populations. In the homologous vector vaccination, the main CD8+ T-cell targets were p7 and NS2, while the NS3 was the target of the DNA/vector-based heterologous vaccination. Moreover, cellular and humoral responses to core, E1, E2, and NS4 were also detected, and the heterologous vaccination induced broader responses compared to the homologous vacine.

These studies demonstrate the effectiveness of various vector platforms (MVA, SFV, and Ad5) and vaccination strategies (homologous and heterologous prime/boost) in eliciting robust immune responses against HCV in animal models. MVA and Ad5 were particularly effective in promoting a Th1-biased immune response, characterized by high IFN-γ production, while SFV excelled in generating strong E2-specific T-cell responses. The heterologous prime-boost strategy using an MVA vector encoding the nearly full-length HCV genome proved to be superior in stimulating a broader immune response across multiple HCV antigens (core, E1, E2, NS3, NS4) compared to the homologous vaccination approaches. Crucially, the studies highlighted the indispensable role of CD4+ T cells in maintaining and expanding the CD8+ T-cell populations, underlining the importance of engaging both cell types for a durable and effective vaccine response. These insights are valuable for tailoring HCV vaccine development to maximize the efficacy and longevity of immune protection.

## 5. RNA-Based Vaccines

### Envelope Glycoprotein 1 and 2

An mRNA construct based on the E1 (amino acids 193–351), E2 (amino acids 386–660), and a modified E2 (with an insertion of a glycosylation site, F442NYT) genes of genotype 1a were encapsulated into lipid nanoparticles (LNPs) and used to vaccinate mice [88]. Following the assessment of the cellular response, the E2_F442NYT_ mRNA-LNP induced the strongest Th1-specific protective immune response (with a significant increase in IL-2 and IFN-γ levels, and granzyme B expression), although the E1_193–351_ showed enhanced IL-2 and IFN-γ levels. However, E1_193–351_/E2_386–660_ and E2_386–660_ elicited a stronger Th2 response, characterized by higher IL-4 and IL-10 levels. Finally, when combining E1_193–351_ and E2_F442NYT_, a better cellular immune response was observed (with a significant increase in the IL-2, IFN-γ, and granzyme B levels).

The study revealed distinct immune responses based on the specific modifications and combinations of the HCV E1 and E2 gene segments. Notably, the modified E2 construct (E2_F442NYT_) significantly enhanced the Th1-specific immune response, characterized by elevated levels of IL-2, IFN-γ, and granzyme B, highlighting the potential of targeted glycosylation modifications to boost cellular immunity. In contrast, the unmodified E1 and E2 constructs primarily elicited a Th2 response, with higher levels of IL-4 and IL-10. Interestingly, combining the E1 segment with the modified E2 construct (E1_193–351_ and E2_F442NYT_) synergistically improved the cellular immune response, suggesting that strategic combinations of mRNA constructs can be more effective in inducing a more comprehensive T-cell response. These findings demonstrate the versatility and potential of mRNA-LNP vaccines to tailor and enhance immune responses against viral antigens, offering valuable insights for the design of more effective HCV vaccines.

## 6. Human Clinical Trials

### 6.1. Clinical Trials Involving Viral Vectors

#### NS3 and NS5B

Two different studies by Swadling and colleagues assessed the immunogenicity of full-length NS3 and NS5B. In the first work, the ChAd3 and MVA vectors were again used with the genetically inactivated NS5B (genotype 1b) polymerase (NSmut) [89]. All volunteers responded to the ChAd3/NSmut prime, with HCV-specific T-cell responses enhanced by the MVA/NSmut boost being more sustained over time, and significantly greater at the end of the study. No HCV-specific T-cell response was detected after MVA/NSmut prime alone. Even if at a lower magnitude, cross reactivity with genotypes 1a, 3a, and 4a responses was generated. The MVA/NSmut boosting vaccination induced higher numbers of both T-cell subsets compared to those seen post ChAd3/NSmut prime. HCV-specific CD4+ and CD8+ T cells were polyfunctional, where the CD4+ T cells produced one (IL-2 or IFN-γ), two (IL-2 and IFN-γ or IFN-γ and TNF-α), or three (IL-2, IFN-γ, and TNF-α) cytokines. The second human clinical trial, using the same constructs, was able to elicit HCV-specific T-cell responses by priming with ChAd3/NSmut and boosting with MVA/NSmut [90]. However, this response was observed in all healthy volunteers, but not in HCV-infected patients, with a lower breadth and magnitude of response. Also, for these patients, boosting with the Ad6/NSmut or MVA/NSmut did not increase the immunogenicity. 

### 6.2. NS3, NS4A, NS4B, NS5A, and NS5B

In the clinical trial with healthy volunteers described by Kelly and colleagues, the ChAd3 and the human adenovirus serotype 6 (Ad6) vectors, with a genetically inactivated NS5B polymerase (NSmut) and expressing the genotype 1b NS3-NS5B, were used [91]. The vaccination was able to elicit cross-reactive T cells directed against immunodominant epitopes (NS3_1406_ KLSGLGINAV, NS3_1073_ CVNGVCWTV, and NS3_1436_ ATDALMTGY), although two NS3_1406_ peptide variants (i.e., 95I and 95B, with a high prevalence at a population level) had a reduced T-cell receptor binding affinity. In the second human clinical trial by the same group, chronically HCV genotype 1-infected patients were immunized with the previously described vaccine constructs using a ChAd3/NSmut prime and a Ad6/NSmut boost, in different regimens, where the volunteers were enrolled in different arms, each one with different prime/boost schedules and vaccine doses [92]. When assessing the vaccination response in the group receiving the PEG-IFN-α/RIB treatment, no responses were seen in the groups receiving the low and medium vaccine doses (5 × 10^8^ and 5 × 10^9^ viral particles (vp)). At the highest dose (2.5 × 10^10^ vp), HCV-specific IFN-γ-secreting T cells were detected in 8/12 (67%) individuals receiving a single prime/boost regimen, whereas double-prime vaccination induced a response in 4/8 (50%) vaccinees and did not significantly enhance the response. However, high responders were observed in the single prime groups. Overall, there was a significant increase in the HCV T-cell response peak after boost but not after priming vaccination. Next, in the untreated group, at the low and medium doses, 2/4 (50%) patients responded to vaccination, while at the higher dose, T cells were induced in 3/4 (75%) patients. There was no effect of vaccination on HCV viral load. Furthermore, the magnitude of the response in the HCV-infected patients was significantly lower, compared to healthy volunteers. The vaccine was able to induce predominantly CD8+ T cells (with low IFN-γ, IL-2, and TNF-α production for the HCV-infected patients).

In a different human clinical trial conducted by Page and colleagues, healthy HCV-uninfected adults who made use of injection drugs within 90 days of receiving the vaccine composed by the ChAd3/NSmut and MVA/NSmut expressing the NS3, NS4, NS5A, and NS5B (genotype 1b) were enrolled [93]. No evidence of vaccine efficacy was detected in the group receiving one vaccine dose (202 participants), where 14 individuals became chronically infected after 6 months. Furthermore, for the group receiving two doses, 19 individuals became chronically infected after 6 months. Despite this outcome, IFN-γ-producing T cells were detected in 78% of vaccinated individuals, although showing a low response.

The vaccination using viral vectors in the described clinical trials proved its efficiency in eliciting a T-cell response. In the work by Kelly and colleagues, it is possible to notice that the highest virus concentration (2.5 × 10^10^ vp) provided better results in terms of IFN-γ-secreting T cells, for either treated or non-treated groups [92]. After the vaccination, no difference in viral load was observed for either group. The vaccine described by Swadling and colleagues was equally effective, eliciting a sustained cross-reactive T-cell response with a polyfunctional profile. However, in their second study, the T-cell response was induced in all healthy individuals, but not in the HCV-infected ones [89]. An explanation for that is T-cell induction impairment by the presence of high viral titers or cellular immune response attenuation following PEG-IFN-α and ribavirin treatment [94,95]. Furthermore, is important to highlight that eliciting an Ab response, particularly targeting the surface glycoproteins (i.e., the structural proteins), is crucial for facilitating the protection and clearance of the infection. On the other hand, Page and colleagues did not observe a protective efficacy in their vaccine study, probably due to the lower vaccine immunogenicity in injection drug users [93,96].

### 6.3. Intramuscular Electroporation for HCV Vaccine Delivery in Human Clinical Trials

#### 6.3.1. NS3 and NS4A

Weiland and colleagues assessed the T-cell response against NS3/4A (of genotype 1a) in a human clinical trial and observed that the IFN-γ-producing T cells increased after two vaccinations, with no improvement with additional boosts [97]. According to the authors, the vaccine had a suboptimal immunogenicity. Therefore, repeated vaccinations may have no beneficial effects. Also, in patients carrying a chronic HCV infection it is likely that the strong regulatory effects are difficult to overcome by vaccination, and four doses were not sufficient to achieve a sustained virological response.

#### 6.3.2. NS3, NS4A, NS4B, and NS5A

A second clinical trial, conducted by Han and colleagues, using a DNA construct based on NS3/NS4A, NS4B, NS5A (of genotypes 1a and 1b) with the IFNL3 gene adjuvant enhanced a vaccine-induced, virus-specific T-cell response in patients with chronic HCV infection [98]. An HCV-specific IFN-γ-producing T-cell response increased by the boost was observed, but not with just a prime vaccination regimen, although with no reduction of HCV RNA titers. An increased IFN-γ, IL-2, TNF, IL-17A production by both CD4+ and CD8+ T cells was observed at week 40, and the T cell frequency was reduced in patients with chronic HCV infection, being inversely correlated with IFN-γ-producing cell levels.

Jacobson and colleagues conducted a human clinical trial with individuals with a history of chronic genotype 1 HCV infection to assess the safety and immunogenicity of their DNA vaccine construct encoding NS3, NS4A, NS4B, and NS5A (genotype 1a/1b) (named INO-8000) alone or co-administered with DNA encoding the IL-12 (named INO-9012) [99]. The individuals were enrolled in different groups: Dose Level 0—receiving 6 mg of the INO-8000 alone, while the remaining three dose-level cohorts received the same INO-8000 dose (6 mg) in combination with different doses of INO-9012 (Dose Level 1—0.3 mg, Dose Level 2—1.0 mg, and Dose Level 3—3.0 mg) at weeks 0, 4, 12, and 24. Immunogenicity was particularly evident when higher doses of IL-12 DNA (dose levels 2 and 3) were co-administered with INO-8000. Cellular responses were induced to all HCV antigens encoded by INO-8000 (NS5A, NS3/4A, NS4B), with the strongest responses to NS5A and the weakest to NS4B. A higher magnitude of HCV-specific CD4+ and CD8+ T-lymphocyte responses was observed, with 75% (15/20) of participants overall, and 100% (5/5) in the dose level 2 cohort, showing increased HCV-specific IFN-γ production. INO-8000 treatment led to increases in the frequency of activated and proliferative CD4+ and CD8+ T cells across all dose levels, with the greatest increases at dose level 2, indicating a potential for targeting and killing HCV-infected cells.

Human clinical trials are the most reliable source to assess vaccine efficacy. Unfortunately, neither of the two studies described here elicited a T-cell response leading to a viral infection resolution. The results suggest that a more complex DNA construct (e.g., NS3/NS4A, NS4B, NS5A) elicited an enhanced cellular immunity. However, the first and third studies assessed only the IFN-γ profile, making it difficult to conclude if the first construct was able to induce a similar cytokine profile. Combining the HCV structural proteins with the NS proteins in these DNA vaccine formulations might enhance not only the cellular immunity but also the humoral response. Furthermore, viral vectors might be an ideal approach to elicit a more robust response and should be considered for future clinical trials.

## 7. Discussion and Future Directions

The above discussed studies shed light on the intricate interplay between HCV structural and NS proteins and the elicited T-cell immune response. The versatility and efficacy of the expression systems, including bacterial and mammalian cells, in producing recombinant proteins for the vaccination studies described above are demonstrated by their ability to express these antigens. Furthermore, the use of safer and more effective innovative vaccination strategies, like VLP-based vaccines, deserves consideration for further development. All these features highlight the significance of these technological advancements in the field.

The use of truncated and soluble versions of HCV antigens demonstrates their immunogenicity as being able to elicit a robust immune response characterized by the induction of both Th1 and Th2 cytokines and showing their potential as vaccine candidates. Moreover, the studies that include adjuvants in the vaccine formulations highlight the importance of the correct adjuvant selection in modulating the immune response, as also evidenced in the shifts from Th1 to Th2 responses based on the used adjuvant. The adjuvant selection is also very important when considering the target population to be vaccinated, e.g., HCV chronically infected patients vs. HCV naïve subjects. In fact, the first category may be affected by immune regulatory mechanisms that need to be overcome for conferring an effective immune response.

As evidenced in the different studies described in this review, HCV NS proteins primarily elicit a T-cell mediated immunity. Unlike the structural proteins, the NS proteins primarily trigger T-cell responses, making them valuable targets for vaccine development [14]. Additionally, as shown in the studies employing different viral vectors and expression systems, NS proteins also demonstrated their role in inducing a T-cell response across different HCV genotypes. Furthermore, the combined use of the structural and NS proteins underlines the synergistic effects of multi-antigenic vaccines. This combination elicited potent CD4+ and CD8+ T-cell responses, suggesting enhanced immunogenicity compared to single antigen-based vaccines. Additionally, studies evaluating novel vaccine constructs, such as in silico-designed mosaic proteins, highlight the potential of enhancing the vaccine efficacy using these approaches. In fact, the mosaic protein-based vaccines are designed to enhance immune responses by displaying multiple viral antigens on a single nanoparticle platform. These vaccines leverage the ability to present a variety of epitopes simultaneously, which can stimulate a robust and broad immune response, potentially offering better protection compared to traditional vaccines. This approach has been proposed and reported for therapeutic HIV, influenza A, and SARS-CoV-2 virus vaccine candidates [100,101,102].

Recently, the increasing interest on nucleic acid-based vaccination strategies is expanding, especially when considering the fact that both DNA and RNA vaccines offer distinct advantages over traditional vaccine platforms. DNA vaccinations, in particular when targeting HCV NS proteins, were able to elicit a robust T-cell response, characterized by IFN-γ secretion and CTL activity. The studies described in this review highlight the versatility and efficacy of DNA vaccines in stimulating both cellular and humoral immunity. On the other hand, HCV RNA vaccines targeting the structural proteins induced a strong Th1-specific immune response, characterized by increased IFN-γ levels and granzyme B expression. Furthermore, combinations of mRNA vaccines targeting different epitopes showed enhanced cellular immune responses compared to single epitope-based vaccines. These findings also demonstrate the potential of mRNA vaccines in eliciting robust and protective immune responses against HCV. An important consideration is the possible use of different adjuvants and its relevance on vaccine formulations in further enhancing the immunogenicity for both DNA and RNA vaccines.

Table 1 shows that all papers described here focused mainly on the IFN-γ (100% of papers assessing the cytokines production), IL-2 (37%), and TNF-α (36%) cytokine profile assessments. IFN-γ is primarily produced by T cells and natural killer (NK) cells, playing a crucial role in mediating, regulating immunity and inflammation, and being a key to the activation of macrophages [103]. IFN-γ enhances the antigen presentation capabilities of these cells, thereby boosting the immune response, also promoting the differentiation of T cells into Th1 cells. This differentiation is critical in shaping the immune response to be more effective against intracellular pathogens. Furthermore, IFN-γ enhances the activation and proliferation of CD4+ and CD8+ T cells, which are essential for controlling viral replication and achieving viral clearance. Elevated levels of IFN-γ are associated with reduced HCV titers and improved liver enzyme levels, suggesting effective viral control [104,105]. Notably, this cytokine inhibits HCV infection by downregulating claudin-1 and CD81, which are critical for the virus’s entry into hepatocytes [106], reducing the susceptibility of liver cells to HCV infection, reinforcing the role of IFN-γ as a protective factor against the virus. Additionally, IFN-γ has antiviral and antitumor properties and is involved in upregulating the expression of the MHC on cells, thereby facilitating the recognition of infected or transformed cells by the immune system.

IL-2 is known primarily for its role in the growth, proliferation, and survival of T cells and is mainly secreted by CD4+ T cells [107]. Dendritic and NK cells can also secrete IL-2 upon activation. Similar to IFN-γ, IL-2 is important for the proliferation and activation of T cells, particularly CD4+ and CD8+ T lymphocytes. Additionally, the incorporation of IL-2 in vaccine strategies could enhance immune activation and improve the overall efficacy of HCV vaccination efforts, providing stronger immunity against the virus [108]. Some studies have indicated that while IL-2 gene expression may be undetectable in some chronic hepatitis C patients, those with acute hepatitis often display strong signals for IFN-γ as a part of their immune response [109]. This suggests that, while IL-2 is a critical cytokine for immune activation, its detectable production may differ significantly between the stages of hepatitis C infection. Nonetheless, given the role of IL-2 in boosting T-cell responses, it has been suggested that therapies enhancing IL-2 production may benefit patients suffering from chronic hepatitis C. Increasing IL-2 levels can potentially improve the functionality of T cells and support their capacity to eliminate HCV-infected cells. This approach necessitates further clinical exploration to evaluate its efficacy and safety as a therapeutic strategy in HCV management.

Finally, TNF-α is a pro-inflammatory cytokine produced by various immune cells, including macrophages, lymphocytes, and NK cells [110]. It is known for its role in the regulation of immune cells and for inducing fever, apoptotic cell death, and inflammation. The induction of TNF-α can significantly enhance the body’s ability to control HCV, ultimately contributing to antiviral responses within the liver [111]. Initially, HCV can induce TNF-α expression rapidly in hepatocytes via toll-like receptors 7 and 8. This induction can be blocked by Abs targeting the E2 protein, highlighting the specific interactions between the virus and host immune mechanisms. The subsequent phases of TNF-α expression are dependent on viral gene expression and replication. Patients with chronic HCV infection exhibit significantly elevated serum levels of TNF-α compared to healthy individuals. These higher levels correlate with the severity of liver inflammation, indicating that TNF-α may serve as a biomarker for assessing disease progression in HCV-related liver conditions. Furthermore, some studies have shown that the presence of TNF-α in circulation is associated with other markers of liver damage, suggesting its role in reflecting hepatic dysfunction [112,113]. Thus, targeting TNF-α presents potential therapeutic avenues for enhancing HCV treatment outcomes.

In the context of HCV vaccine development, IL-4 and IL-10 are also important and were sometimes assessed by these studies. IL-4 is involved in the differentiation of naïve T cells into Th2 cells, which are crucial in the activation of humoral immunity [114]. It helps in the production of Abs by B cells, thus contributing to a balanced immune response that targets HCV. Elevated serum levels of IL-4 have been documented in a subset of patients with chronic HCV infection, suggesting a modulation of the immune response that may affect the body’s ability to control the virus [115]. Despite its elevated levels in circulation, IL-4 production in the liver appears limited, indicating that its systemic effects may not translate to local immune activation within the liver. Targeting IL-4 presents potential therapeutic avenues in managing HCV infection. Increasing IL-4 levels may promote collagen synthesis and influence fibrogenesis, particularly in the context of liver fibrosis associated with chronic HCV [115]. Furthermore, recent studies suggest that incorporating IL-4 levels into diagnostic models may improve predictive outcomes for liver fibrosis in HCV-infected patients, demonstrating the cytokine’s clinical significance [116,117]. On the other hand, IL-10 is primarily known for its potent anti-inflammatory properties, playing a crucial role in modulating inflammation and maintaining cell homeostasis, being expressed and produced by various immune cells, including CD4+ T cells, macrophages, and dendritic cells, among others [118]. In HCV infection, elevated levels of IL-10 can decrease effector T-cell responses, promoting viral persistence and complicating disease management [119]. Some studies have reported that patients with chronic HCV often exhibit increased circulating IL-10 levels. In a cohort study, no significant difference in IL-10 levels was found between healthy controls and overall patients with chronic hepatitis C; however, those with more severe necroinflammation (auto-amplification loop driven by necrosis and inflammation) showed significantly elevated IL-10 levels [120]. This suggests that increased IL-10 levels may correlate with disease severity and ongoing inflammation in the liver. There is potential for targeting IL-10 to enhance therapeutic strategies for HCV infection. The neutralization of IL-10, particularly the IL-10 produced by regulatory B cells, has shown promise in experimental models, suggesting that reducing IL-10 levels may bolster antiviral immunity and promote better clinical outcomes in HCV patients [121]. However, achieving a balance is essential, as excessive suppression of IL-10 could result in increased inflammation and tissue damage.

When the majority of the papers included in this review focused on the T-cell response, the cytokines related to the humoral response regulation were not prioritized. Noteworthy, only one paper evaluated the IL-18 production (Table 1) involved in innate and adaptative immune responses, acting on Th1 cells, macrophages, natural killer cells, B cells, and dendritic cells [122]. Elevated levels of IL-18 have been observed in patients with chronic HCV infection, suggesting a potential role in viral pathogenesis. Moreover, serum concentrations of IL-18 are significantly higher in HCV-positive individuals compared to healthy controls, indicating its involvement in the disease process [123]. The assessment of interleukins provides essential insights into T-cell immunity, revealing the complex network of cytokine interactions that regulate immune responses. This assessment is crucial for developing targeted therapies and understanding the overall dynamics of the immune system. As our knowledge expands, so does the potential for leveraging interleukin assessments in personalized medicine, where cytokine profiles could guide individualized treatment plans for a range of immunological diseases.

It is noteworthy that, for the protein/peptide-based vaccine, E2 and NS3 were the most used proteins included in the vaccine formulation, while for the DNA-based vaccines, NS3, followed by core and E2, were extensively used (Figure 2). The E2 and NS3 proteins are essential for the virus life cycle, thereby rendering them important targets for vaccine development. E2 is essential for viral entry into the host cells, mediating attachment to cellular receptors and subsequent fusion of the viral and cellular membranes (Figure 1). Furthermore, E2 interacts with host cell receptors such as CD81, scavenger receptor class B type I, and the tight junction proteins claudin-1 and occluding [124]. Consequently, E2 is an essential component for HCV’s attachment and penetration of host liver cells. Its pivotal role in the viral entry process justifies the focus on E2 as a primary target for vaccine strategies. By targeting E2, vaccines can potentially elicit neutralizing Abs (nAbs) that block this critical interaction, thereby preventing infection [125]. Despite its variability, certain conserved regions of E2 are critical for its function and are targeted by broadly neutralizing antibodies (bnAbs) [126]. On the other hand, NS3 is a multifunctional protein with serine protease and helicase activities, essential for the replication of HCV (Figure 1). The protein role in viral replication makes it a critical component of the viral machinery, which means it is less tolerant to mutations. This multifunctionality signifies NS3’s importance as a target for vaccine development, as effectively inhibiting its function could impede HCV replication. Furthermore, NS3 is a major target of the cellular immune response, particularly CD8+ T cells. Effective T-cell responses against NS3 are associated with the control of HCV infection and spontaneous clearance of the virus [127]. In fact, it has been one of the main targets of first-generation HCV-specific antiviral drugs (e.g., Telaprevir and Boceprevir).

The importance of the B-cell response in the context of HCV infection and vaccine development is also critical, providing a complementary defense mechanism to the T-cell-mediated response. During a natural HCV infection, the B-cell response is primarily responsible for producing nAbs, mainly targeting epitopes on the viral envelope proteins E1 and E2 [126]. Some studies have shown that, during the early acute phase of infection, there is a reduction in the number of circulating B cells [128], whereas in the chronic infection, patients typically exhibit elevated levels of interferon-stimulated genes and B-cell activation [129,130,131]. This activation is characterized by an increased expression of activation markers, including CD69, CD71, CD86, and the chemokine receptor CXCR3. Furthermore, there is evidence of B-cell infiltration in the liver during chronic infection [132,133]. In the acute phase of HCV infection, there is a rapid induction of nAbs, which decline or disappear as the infection resolves. In the early phase of chronic HCV infection, these Ab titers are low or even absent, and the production of nAbs in the late chronic phase is insufficient to eliminate the viral quasi-species [134]. However, the isolation of bnAbs has been documented in patients who undergo spontaneous HCV clearance as well as in chronically infected individuals [135,136]. Numerous studies have demonstrated the effectiveness of bnAbs as potential prophylactic and therapeutic tools, showing their capability to confer protection and contribute to viral clearance in various in vitro and in vivo models [137,138,139,140]. However, the high HCV genetic diversity poses a significant challenge, allowing the virus to escape the immune surveillance and persist in the host. Despite these difficulties, the bnAbs can neutralize a range of HCV genotypes and play a crucial role in controlling the infection [134,141]. Inducing nAbs responses through vaccination is considered a priority in HCV vaccine development, and several bnAbs have been isolated from human subjects using different methods such as B-cell immortalization, phage and yeast-Ab display, as well as from immunized animals [142,143,144].

The ability of the virus to induce chronic infection in many individuals has been partly attributed to its capability to evade the humoral immune response. This chronic immune stimulation is often marked by the occurrence of a variety of autoantibodies, such as anti-nuclear Abs (10–40% of infected patients), anti-smooth muscle Abs (7–27%), and rheumatoid factor (RF) (45–70%) [145]. Several studies showed that the autoantibodies in HCV-infected patients can have a significant systemic impact, leading to conditions such as polyarthralgia (40–80% of infected patients), arthritis (4–5%), sicca syndrome (20–30%), and cryoglobulinemia vasculitis (5–40%) [146]. According to Priora and colleagues, different evidences indicate that at least one serum immunological abnormality is present in up to 50% of HCV-infected patients. The mechanism underlying cryoglobulin formation and mixed cryoglobulinemia remains poorly understood; however, it is associated with B-cell dysregulation and the production of RF [126]. The reasons behind the frequent induction of RF production following HCV infection are not entirely clear [147,148,149]. Additionally, it is uncertain how HCV-induced RF generates immune complexes that are susceptible to precipitation under hypothermic conditions. In recent years, numerous studies have focused on various Abs and their relationships with chronic hepatitis C. Furthermore, these autoantibodies may indicate the immune system’s effort to clear the virus or may accompany clinical autoimmune features. Persistent viral stimulation leads to a polyclonal expansion of B cells, resulting in the formation of immune complexes, giving rise to a wide spectrum of autoimmune and lymphoproliferative disorders, which may present as clinical conditions, serological markers, or both [150,151,152,153,154]. The introduction of the DAAs has notably improved or resolved many liver-related and extrahepatic complications, including rheumatic ones, but further investigations are required to understand the clinical outcomes based on the serological profiles [150,155].

When evaluating the safety profile of HCV vaccine candidates, several key aspects need to be addressed to ensure that the vaccine is both effective and safe for widespread use. These include monitoring for adverse reactions, long-term safety, and specific considerations for different population groups [156]. As mentioned above, one possible severe adverse event may be the occurrence of autoimmune reactions. It is crucial to monitor for potential autoimmune responses triggered by the vaccine, including any new onset of autoimmune diseases or exacerbation of existing conditions. The vaccine safety profile should be evaluated across different age groups, including children, adults, and the elderly, to ensure its safety for all populations. Moreover, individuals with preexisting health conditions, such as liver disease, HIV infection, or other chronic illnesses, should be included in the safety assessments to ensure the vaccine does not exacerbate their conditions. The immune response generated by the vaccine should be strong enough to confer protection against HCV while maintaining a favorable safety profile. This balance is critical to minimize the risk of adverse effects while achieving an effective immunization.

In recent clinical trials, HCV regimens have demonstrated an acceptable safety profile, with no serious adverse events reported [157]. Regarding the human clinical trials included in this review, Kelly and colleagues did not reported data concerning vaccine safety [91]. On the other hand, in their subsequent clinical trial, mild local and systemic side effects were reported, with the vaccination being well tolerated, with no vaccine-related serious adverse events [92]. The vaccine used in the clinical trial by Swadling and colleagues was reported to have an acceptable safety profile [89]. No severe adverse reactions were reported in the study conducted by the same group, where all localized adverse reactions were of mild severity, with local pain occurring in 25% of vaccinees and few systemic adverse reactions being observed, with the most common being myalgia, malaise, and headache in 16.67% of vaccinees [90]. Finally, Page and colleagues reported no vaccine-related serious adverse events [93]. The most frequent laboratory adverse event was the elevation of the alanine aminotransferase level, a finding known to be associated with substance abuse as well as HCV infection. Comprehensive safety data are essential for encouraging public and healthcare provider confidence in vaccine efficacy and safety. The lack of serious adverse reactions during clinical trials is significant for prospective vaccine recipients, especially considering that no effective vaccine against HCV is currently available to the public. Ongoing monitoring of vaccine safety will be necessary as clinical trials progress and vaccinations begin in broader populations. The use of different systems for reporting adverse events will help in assessing the long-term safety of HCV vaccines as they become available to the public, ensuring that any emerging safety issues are promptly addressed.

In regards to the other NS proteins (i.e., p7, NS2, NS4, and NS5) and the core protein, the protein/peptide- and DNA-based vaccines used these as targets in a similar proportion (Figure 2). Only 17% of the DNA-based vaccines used E1, compared to 41% of the protein/peptide-based vaccines. An effective HCV vaccine would likely need to target multiple viral components to elicit a broad and robust immune response. Combining E2 and NS3 (or even the other NS proteins) in a single vaccine formulation could offer several advantages: (i) by targeting E2, the vaccine could induce nAbs to prevent infection, while NS3 could stimulate strong T-cell responses to eliminate infected cells; (ii) a multi-antigen approach can reduce the likelihood of viral escape through mutations, as the virus would need to simultaneously evade both the Ab and T-cell responses; and (iii) using advanced vaccine platforms, such as VLPs, DNA vaccines, or viral vectors, can enhance the presentation of these antigens to the immune system, improving the overall immunogenicity and protective efficacy.

The HCV genetic variability also hampers the development of a vaccine that can effectively induce bnAbs across different HCV strains and genotypes. However, understanding the mechanisms of immune escape and the structural bases of neutralization can aid in the design of immunogens that elicit more potent and bnAbs responses. Recent vaccine research has increasingly focused on designing envelope protein-based vaccines that aim to elicit robust Ab responses. For instance, engineered versions of the E2 protein and other viral antigens have been used to enhance the immunogenicity and stability of vaccine candidates, attempting to focus the immune response on conserved epitopes [158,159,160]. Moreover, combining B-cell epitope-based vaccines with T-cell vaccines is being explored to provide more comprehensive immune protection, targeting both early infection stages through nAbs and clearing infected cells via cellular immunity. The B-cell response plays a pivotal role in controlling HCV infection and is central to the development of an effective HCV vaccine. Advances in our understanding of the humoral immune response to HCV and its evasion strategies are critical in guiding the development of next-generation vaccines that can prevent the initial infection or reduce the virus’ ability to establish a chronic infection.

In this review, the majority of studies have concentrated their focus on the HCV genotype 1 (Figure 3). Furthermore, this trend is evident in the field of HCV research, in part due to the fact that early discoveries and characterizations of HCV molecular biology involved these genotypes. As a result, initial diagnostic tools, therapeutic regimens, and research methodologies were developed with a focus on genotypes 1a and 1b, setting a precedent for subsequent studies. Other than that, these genotypes are the most prevalent globally, particularly in North America and Europe [161]. This high prevalence has driven research interest and funding toward these genotypes to address the larger public health burden they represent.

While this focus has yielded valuable insights and advancements, it also presents certain limitations that need to be addressed for a more comprehensive understanding of HCV and the development of universally and broadly effective interventions. There are at least eight major HCV genotypes, each with multiple subtypes. Genotypes 2 through 8 (more common in low- and middle-income countries) are less studied (particularly in the case of the DNA-based vaccines, Figure 3), leading to gaps in knowledge about their epidemiology, pathogenesis, and response to treatment. The lack of research on these genotypes can exacerbate health disparities, as populations in these regions may not benefit from advancements primarily tailored to genotypes 1a and 1b. Also, HCV is highly variable, and the genetic diversity among different genotypes can influence the virus infection mechanisms and biology and its interaction with the host immune system [162]. Focusing predominantly on genotype 1 may overlook critical differences in immune evasion strategies and treatment resistance mechanisms present in other genotypes. An effective HCV vaccine needs to provide broad protection across multiple genotypes. The current focus on genotype 1 might limit the identification of conserved epitopes and immune responses critical for a pan-genotypic vaccine. This focus could delay the development of a vaccine capable of providing universal protection. To overcome these limitations, the research community should prioritize inclusive studies encompassing a broader range of HCV genotypes.

The study of HCV is hindered by the lack of a robust cell culture system or an appropriate animal model, as the virus is highly specific to humans. Additionally, the use of NHPs is limited due to ethical concerns, complicating the assessment of viral clearance. Consequently, evaluating the viral clearance, protection, and survival rate given by the T- and B-cell response is not feasible with the mouse models used in the papers described here. Although mice with humanized livers can serve as a murine model for HCV infection, generating and maintaining these mice is technically challenging and expensive [163]. Furthermore, the papers reviewed did not utilize these models, and they do not characterize a mortality rate following infection.

While HCV treatments has advanced considerably in the last decade, the absence of a preventive vaccine still represents a gap in the fight against HCV. However, the advancements in the knowledge of HCV biology and antiviral mechanisms offer promising avenues for the future development of an effective HCV vaccine. One important aspect that will definitely help in developing an effective prophylactic countermeasure for HCV is the understanding of the intricate dynamics of the virus and the human immune system. HCV has the ability to evade immune detection mechanisms, leading to chronic infections in the majority of infected subjects. Therefore, the current research is focusing on elucidating these mechanisms and identifying viral vulnerable points that could be targeted by new vaccine strategies. Moreover, the integration of vaccine development with the implementation and improvement of other preventive strategies, especially in low-income countries, such as screening for blood transfusions and assistance for injection drug users, can be essential for comprehensive HCV control efforts. While challenges persist, the future of HCV vaccine development may be not too far. Through continued scientific innovation, technological advancements, and collaborative efforts, the goal of HCV control and elimination by 2030 represents a reachable goal.

## 8. Conclusions

The results discussed in this review contribute with valuable insights into the development of effective HCV vaccines, emphasizing the importance of understanding the immune response to viral antigens and the rational design of vaccine candidates. These studies are crucial for future research aimed at advancing vaccine strategies against HCV, with implications for global health and biopharmaceutical innovation.

## Figures and Tables

**Figure 1 vaccines-12-00890-f001:**
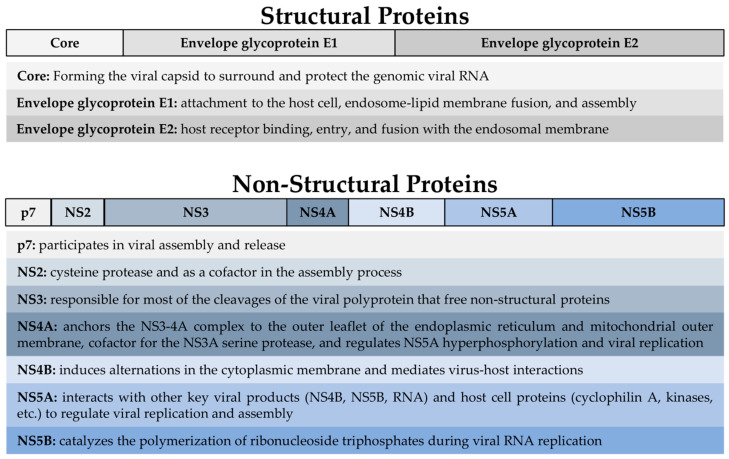
The HCV genomic organization and translation products. The HCV genome consists of a positive single-stranded RNA genome (~9.6 kb) encoding a polyprotein. This polyprotein is cleaved by cellular and viral proteases into ten different proteins, the structural and NS proteins, as detailed in the schematic representation. The structural and NS proteins functions are also described.

**Figure 2 vaccines-12-00890-f002:**
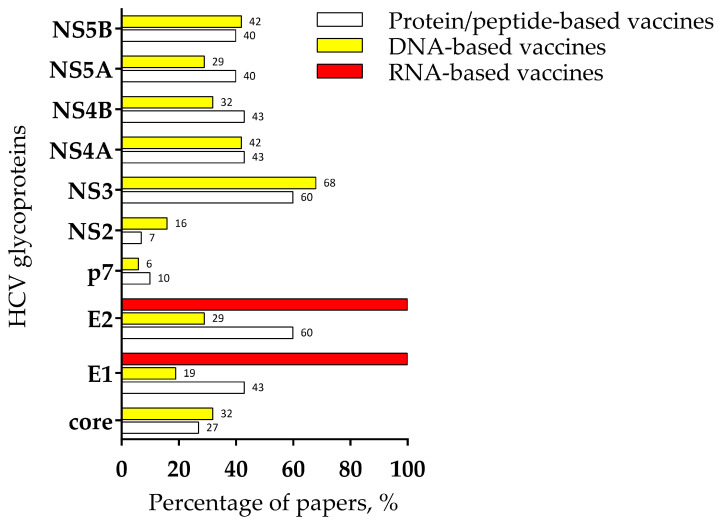
Frequency of HCV glycoproteins used as vaccine targets from the papers described in this review. The white, yellow, and red bars represent the protein/peptide-, DNA-, and RNA-based vaccines, respectively. The bars are followed by the protein utilization frequency by each approach.

**Figure 3 vaccines-12-00890-f003:**
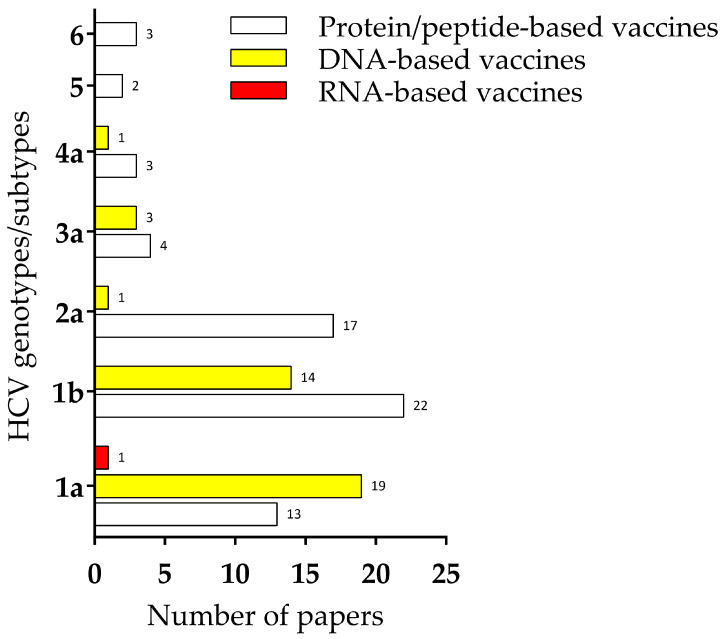
HCV genotypes and subtypes that are the focus of the reported studies and enumerated based on the vaccine strategy utilized. The white, yellow, and red bars represent the protein/peptide-, DNA-, and RNA-based vaccines, respectively. The absolute number of papers using each genotype/subtype is indicated by the corresponding bars. When not specified by the authors, HCV genotype 1 was considered as 1a. The same study may have used different genotypes/subtypes in their vaccine formulation.

**Table 1 vaccines-12-00890-t001:** Description of the cytokines assessed by each study. The cytokines reported in each study are listed below, regardless of the assay and cell type used. The studies not reporting cytokines assessment were omitted (e.g., studies reporting CTL only).

Reference	Cytokines									Cellular Response
Protein-Based Vaccines										
[26]	IFN-γ					IL-4				
[27]	IFN-γ					IL-4				
[28]	IFN-γ					IL-4				
[30]	IFN-γ					IL-4				
[40]	IFN-γ		IL-2			IL-4				
[44]	IFN-γ	TNF-α	IL-2	IL-12				IL-6		
[92]	IFN-γ	TNF-α	IL-2				IL-17			granzyme A, granzyme B
[84]	IFN-γ	TNF-α	IL-2							
[42]	IFN-γ	TNF-α	IL-2							
[91]	IFN-γ	TNF-α	IL-2							
[90]	IFN-γ	TNF-α	IL-2							
[49]	IFN-γ	TNF-α	IL-2							
[36]	IFN-γ	TNF-α	IL-2							
[50]	IFN-γ	TNF-α								
[31]	IFN-γ	TNF-α								
[46]	IFN-γ	TNF-α								
[32]	IFN-γ							IL-6	IL-10	
[41]	IFN-γ								IL-10	
[35]	IFN-γ			IL-12	IL-18					granzyme B
[34]	IFN-γ									granzyme B
[51]	IFN-γ									
[37]	IFN-γ									
[33]	IFN-γ									
[29]	IFN-γ									
[52]	IFN-γ									
[53]	IFN-γ									
[54]	IFN-γ									
[43]	IFN-γ									
[93]	IFN-γ									
DNA-Based Vaccines										
[98]	IFN-γ	TNF-α	IL-2				IL-17			
[79]	IFN-γ	TNF-α	IL-2							granzyme B
[87]	IFN-γ	TNF-α	IL-2							
[71]	IFN-γ	TNF-α	IL-2							
[73]	IFN-γ	TNF-α	IL-2							
[82]	IFN-γ	TNF-α	IL-2							
[80]	IFN-γ	TNF-α	IL-2							
[76]	IFN-γ	TNF-α	IL-2							
[84]	IFN-γ	TNF-α	IL-2							
[83]	IFN-γ	TNF-α				IL-4	IL-17			
[77]	IFN-γ		IL-2							
[70]	IFN-γ		IL-2							
[86]	IFN-γ		IL-2			IL-4				
[59]	IFN-γ					IL-4				granzyme B
[64]	IFN-γ					IL-4				
[81]	IFN-γ					IL-4				
[61]	IFN-γ					IL-4				
[60]	IFN-γ					IL-4				
[67]	IFN-γ					IL-4				
[66]	IFN-γ					IL-4				
[85]	IFN-γ									
[97]	IFN-γ									
[65]	IFN-γ									
[74]	IFN-γ									
[72]	IFN-γ									
[62]	IFN-γ									
[75]	IFN-γ									
[99]	IFN-γ									
RNA-Based Vaccines										
[88]	IFN-γ		IL-2			IL-4			IL-10	granzyme B

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
