# Peer review of "Exploring T-Cell Immunity to Hepatitis C Virus: Insights from Different Vaccine and Antigen Presentation Strategies"

_vaccines, 2024, doi:10.3390/vaccines12080890_

Round 1

Reviewer 1 Report

Comments and Suggestions for Authors

In this review, Dr Costa and Sautto addressed the importance of HCV proteins and vaccination strategies in vaccine development. They reviewd the literature studies exploring the relationship between HCV structural and non-structural (NS) proteins and their role in contributing to the elicitation of an effective T-cell immune response and their sequence conservation, making them valuable vaccine targets. They also discussed the use of different vaccine platforms, such as viral vectors and virus-like particles, their versability and efficacy for vaccine development. Of potential clinical impact, they discussed how diverse HCV antigens demonstrated immunogenicity, eliciting a robust immune response, and making them as promising vaccine candidates for either protein/peptide-, DNA- or RNA-based vaccines.

The manuscript is of interest and, in general, it is well-presented. I have only a comment to further improve the manuscript. Other than efficacy, the authors should also discuss the safety profile of vaccine candidates. This is in my opinion a very important issue as HCV infection is significantly related to the development of autoimmunity. In this regard, the authors should recall previous literature studies and pathogenic mechanisms underlying the development of autoimmunity and autoantibodies in HCV patients as summarized in a comprehensive review (HCV and autoimmunity. Curr Pharm Des. 2008;14(17):1678-85. doi: 10.2174/138161208784746824.). Importantly, the development of autoimmunity and serum autoantibodies migh have a non established and controversial  effect and impact on antiviral therapy in HCV patients, as previously reported (Clinical impact of non-organ-specific autoantibodies on the response to combined antiviral treatment in patients with hepatitis C. Clin Infect Dis. 2005 Feb 15;40(4):501-7. doi: 10.1086/427285. ).

Author Response

In this review, Dr Costa and Sautto addressed the importance of HCV proteins and vaccination strategies in vaccine development. They reviewed the literature studies exploring the relationship between HCV structural and non-structural (NS) proteins and their role in contributing to the elicitation of an effective T-cell immune response and their sequence conservation, making them valuable vaccine targets. They also discussed the use of different vaccine platforms, such as viral vectors and virus-like particles, their versability and efficacy for vaccine development. Of potential clinical impact, they discussed how diverse HCV antigens demonstrated immunogenicity, eliciting a robust immune response, and making them as promising vaccine candidates for either protein/peptide-, DNA- or RNA-based vaccines.

The manuscript is of interest and, in general, it is well-presented. I have only a comment to further improve the manuscript. Other than efficacy, the authors should also discuss the safety profile of vaccine candidates. This is in my opinion a very important issue as HCV infection is significantly related to the development of autoimmunity. In this regard, the authors should recall previous literature studies and pathogenic mechanisms underlying the development of autoimmunity and autoantibodies in HCV patients as summarized in a comprehensive review (HCV and autoimmunity. Curr Pharm Des. 2008;14(17):1678-85. doi: 10.2174/138161208784746824.). Importantly, the development of autoimmunity and serum autoantibodies might have a non-established and controversial  effect and impact on antiviral therapy in HCV patients, as previously reported (Clinical impact of non-organ-specific autoantibodies on the response to combined antiviral treatment in patients with hepatitis C. Clin Infect Dis. 2005 Feb 15;40(4):501-7. doi: 10.1086/427285.).

We appreciate the reviewer's suggestion. As recommended, a discussion on autoimmunity associated with HCV infection was included (lines 1,200-1,223). Additionally, we have included a discussion towards the importance of assessing the vaccine safety profile (lines 1,224-1,260).

Reviewer 2 Report

Comments and Suggestions for Authors

Please find below the comments and suggestions for the manuscript entitled "Exploring the T cell Immunity to Hepatitis C Virus: Insights from Different Vaccine and Antigen Presentation Strategies”

While the manuscript is focused on the T cell-based immunity against Hepatitis C virus and covers the topic extensively, the scope of antibody mediated immune response should also be briefly discussed in the context, particularly to emphasize the role of T cell immunity. Only minor mention is given in this regard in the manuscript. A brief discussion (in the Introduction or Discussion section) about the role of antibody mediated responses, and if they do or do not appear to contribute significantly to viral protection or control would be helpful.

Major comment: Throughout the manuscript, while the levels of cytokine production /profiles are mentioned that a candidate protein or vaccine generated, it is not sufficiently discussed what these responses translated to in terms of viral clearance, protection and efficacy (except for some general mention in lines 356, 413, 429, 441, 588, 896 and 902). It is not mentioned what neutralization in vitro assays were used to evaluate the vaccine efficacy and what were the results and outcomes? Or how the vaccine candidates performed in the animal studies, what percentage /how many animals out of the total number of animals remained protected, and for how long, survival rates in each case etc.    

Materials and Methods:

Lines 107-109, 110: The adapted criteria 1-4 are ‘inclusion criteria’ for topic in the current manuscript, not exclusion criteria.

Also, it appears from the condition 1 of the adapted criteria that only open access papers (or those available through the library system) were cited. Does this imply that any relevant references not available through open access (or through library access) would have been missed in this review? Were there any relevant papers that could not be cited in this review due the set inclusion criteria? 

Peptide/protein based vaccines:

Line 131-132: In this context, the production of HCV recombinant proteins using different systems allowed the evaluation of the HCV-elicited immune response and their role in conferring an effective antiviral response.

Please amend as: …. Different systems allowed the evaluation of immune responses against potential HCV vaccine target proteins and their role…

Line 134: Soluble E2 expression using different systems and adjuvants

Amed as: Soluble E2 expression using different systems and their formulation with various adjuvants

Lines 189-190: In section 3.1.2

It appears that non-structural proteins are discussed under ‘Structural proteins’ (section 3.1)?

Line 461: Edit required……….translated to the desired antigen..

Line 678: While non-human primates have been abbreviated earlier in line 141, please mention in full in the title here.

All clinical trial studies can be put under one main heading, followed by subheadings.

Discussion and future directions:

Line 921-925: Please avoid using the word  ‘these’ throughout this paragraph, such as

….these expression systems….

The expression systems should be defined (such as bacterial, mammalian etc) in this new section.

Language correction required in ‘……to be utilized…’

Lines 947-949: Additionally, studies evaluating novel vaccine constructs, such as mosaic proteins designed in silico, highlight the potential of enhancing the vaccine efficacy using these promising approaches. Please clarify/review this statement and provide supportive reference.

Line 961: Review the statement “An important consideration is the possible use of different adjuvants and its relevance on vaccine formulations in further enhancing the immunogenicity for both DNA and RNA vaccines”.

This is contradictory to the statement given in lines 473-475: “Additionally, these vaccines do not require live pathogens or adjuvants, reducing safety concerns associated with traditional vaccine platforms”.

Which adjuvants can be used for DNA and RNA vaccines?

Lone 979: It is mainly produced by CD4+ T cells, but CD8+ T cells. This statement is incomplete.

Line 1003-1005: Review the statement:

It is known that E2 and NS3 play critical roles in the viral life cycle and immune evasion, making them important targets for vaccine development.

Figure 3 legend: Remove repeat words.

All discussion section: The results discussed from respective studies should also include what animal model (or clinical trial) it refers to.    

Other:

Overall, the layout (headings and subheadings) of the manuscript are little difficult to follow for a reader. A clear layout could include listing the structural, non-structural proteins, and a combination of structural + non-structural proteins, and then describing all the vaccination strategies that have been tested for each protein in that category (which could come under different subheadings). In this way both the proteins and the vaccination strategies would be more distinctly laid out. For example below:

(A) Structural proteins:

1. Core

2. Envelope glycoprotein 1

3. Envelope glycoprotein 2

(B) Non-structural proteins:

1. P7

2. NS2

3. NS3

4. NS4A

5. NS4B

6. NS5A

7. NS5B

(C) Combination of Structural and Non-structural proteins

Line 29: Minor edit required

It is crucial to understand that….

Line 1155: Ref 15

Grakoui, A.; Shoukry, N.H.; Woollard, D.J.; Han, J.-H.; Hanson, H.L.; Ghrayeb, J.; Murthy, K.K.; Rice, C.M.; Walker,  C.M. HCV Persistence and Immune Evasion in the Absence of Memory T Cell Help. Science (1979) 2003, 302, 659–662.

Please remove year (1979). The correct year of publication is 2003.

Comments on the Quality of English Language

Minor editing required in some sections (as mentioned in the review report).

Author Response

While the manuscript is focused on the T cell-based immunity against Hepatitis C virus and covers the topic extensively, the scope of antibody mediated immune response should also be briefly discussed in the context, particularly to emphasize the role of T cell immunity. Only minor mention is given in this regard in the manuscript. A brief discussion (in the Introduction or Discussion section) about the role of antibody mediated responses, and if they do or do not appear to contribute significantly to viral protection or control would be helpful.

We appreciate the reviewer's suggestion and a broader discussion regarding the role of the humoral response was included in the Discussion and Future Directions section (lines 1,178-1,199).

Major comment: Throughout the manuscript, while the levels of cytokine production/profiles are mentioned that a candidate protein or vaccine generated, it is not sufficiently discussed what these responses translated to in terms of viral clearance, protection, and efficacy (except for some general mention in lines 356, 413, 429, 441, 588, 896 and 902). It is not mentioned what neutralization in vitro assays were used to evaluate the vaccine efficacy and what were the results and outcomes? Or how the vaccine candidates performed in the animal studies, what percentage/how many animals out of the total number of animals remained protected, and for how long, survival rates in each case etc.

The authors appreciate the reviewer's observation. A broader discussion about the role of the described cytokines during the HCV infection and how they can assist the immune system to confer protection against the virus infection was added (lines 1,051 -1,138).

The study of HCV is hindered by the lack of a robust cell culture system or an appropriate animal model, as the virus is highly specific to humans. Additionally, the use of non-human primates is limited due to ethical concerns, complicating the assessment of viral clearance (we have included a justification regarding this topic in lines 1,326-1,334s). None of the papers included in this review that describe the vaccination of non-human primates (references 28, 37, 44, 79, and 80) reported any survival rates. This is expected, as these models do not typically exhibit a mortality rate following infection. Furthermore, during the studies the animals did not develop adverse effects that would necessitate euthanasia. Instead, these models are primarily characterized by the rate at which chronic infection develops.

While neutralization assays are necessary to evaluate the neutralizing activity of antibodies, which can serve as indirect evidence of their protective capability, the discussion of these assays was not included in this review. This omission is due to the focus of our review on cellular responses, and the inclusion of the studies focusing on the antibody response would fall outside of the scope of our work. However, when reporting the results of studies assessing both the humoral and cellular response, the discussion of these data has been included. The information gathered here pertains to the cellular response elicited by vaccines, as described by assays such as ELISPOT and flow cytometry. These assays detect T-cell secreted cytokines in response to vaccine stimulation. Similar to the humoral response, the lack of an appropriate animal model hampers the assessment of viral clearance and protection by the cellular response.

Additionally, since the animals are not infected, parameters such as survival rates are not reported. In fact,currently mice with a humanized liver can be used as a murine model for HCV infection. However, generating and maintaining mice with humanized livers is technically challenging and expensive and the papers described in our review did not use these models. Additionally, they are not characterized by a mortality rate following the infection.

A different approach to overcome these limitations is to use surrogate challenge models. Out of the sixty-two papers included in this review, only five used these models (references 36, 51, 52, 76 and 77). In the work of Filskov and colleagues, transient transfection of liver cells was achieved through hydrodynamic infection with plasmid DNA encoding the p7 protein. The authors found that vaccinated mice cleared transgene-expressing cells following vaccination (lines 259-262). The other studies employed different surrogate challenge models. Martinez-Donato and colleagues challenged mice with vaccinia virus expressing HCV structural proteins and observed a remarkable reduction in viral titers in vaccinated animals(lines 387-389). In a subsequent study, the same group used vaccinia virus to express either the core protein alone (vvCo) or HCV structural proteins (vvRE) and found that vaccinated mice showed suppression of viremia for both vvCo and vvRE challenges (lines 394-397). In the work of Tan and colleagues, the vaccinated mice were challenged with a murine herpes virus expressing the NS3, and no virus was detected in the spleen, suggesting an effective viral control (lines 671-673). Finally, Fournillier and colleagues used a challenge model based on recombinant Listeria monocytogenes expressing the NS3 protein and observed that vaccinated mice efficiently reduced Listeria titers (lines 678-681).

Materials and Methods:

Lines 107-109, 110: The adapted criteria 1-4 are ‘inclusion criteria’ for topic in the current manuscript, not exclusion criteria. 

We appreciate the reviewer's correction and have made the appropriate modification to the manuscript. (line 111 and 115).

Also, it appears from the condition 1 of the adapted criteria that only open access papers (or those available through the library system) were cited. Does this imply that any relevant references not available through open access (or through library access) would have been missed in this review? Were there any relevant papers that could not be cited in this review due the set inclusion criteria?  

Fortunately, the Cleveland Clinic library system was able to provide access to most of the non-open access papers that were relevant to the review, despite the initial criterion that may have implied their exclusion. This allowed us to thoroughly review the available literature without limiting the scope of the study due to access restrictions. 

Peptide/protein-based vaccines:

Line 131-132: In this context, the production of HCV recombinant proteins using different systems allowed the evaluation of the HCV-elicited immune response and their role in conferring an effective antiviral response. Please amend as: …. Different systems allowed the evaluation of immune responses against potential HCV vaccine target proteins and their role… 

We thankfully acknowledge the reviewer's correction and have made the necessary modifications to the manuscript. The revised version now precisely conveys the addressed concern (lines 137-138).

Line 134: Soluble E2 expression using different systems and adjuvants. Amend as: Soluble E2 expression using different systems and their formulation with various adjuvants

We are appreciative of the reviewer's feedback and have updated the manuscript to address this point. The revised text now properly includes the suggested modification (section 3.1.2).

Lines 189-190: In section 3.1.2. It appears that non-structural proteins are discussed under ‘Structural proteins’ (section 3.1)?

For this particular paper (Donnison et al., 2021) under the ‘Structural protein’ section, after vaccination with the E2, the authors used the mouse splenocytes to ex vivo­-stimulate them using not only a structural proteins peptide pools, but also, non-structural proteins peptide pools. The T-cell response was then assessed by intracellular cytokine staining and flow cytometry. A clarification was added in this particular result discussion (lines 199-202).

Line 461: Edit required……….translated to the desired antigen..

 The authors are appreciative of the reviewer's feedback and have updated the manuscript to address this point (lines 427-428).

Line 678: While non-human primates have been abbreviated earlier in line 141, please mention in full in the title here.

The revised text now includes this suggestion (line 704).

All clinical trial studies can be put under one main heading, followed by subheadings.

The authors sincerely appreciate the reviewer's suggestion. ​Consequently, all human clinical trials have been organized into a new section (6. Human Clinical Trials). This section is further divided into subheadings: “Clinical Trials Involving Viral Vectors” (line 875) and “Intramuscular Electroporation for HCV Vaccine Delivery in Human Clinical Trials” (line 946).

Discussion and future directions:

Line 921-925: Please avoid using the word  ‘these’ throughout this paragraph, such as….these expression systems….The expression systems should be defined (such as bacterial, mammalian etc) in this new section. Language correction required in ‘……to be utilized…’

The revised text now includes the suggested corrections (line 997).

Lines 947-949: Additionally, studies evaluating novel vaccine constructs, such as mosaic proteins designed in silico, highlight the potential of enhancing the vaccine efficacy using these promising approaches. Please clarify/review this statement and provide supportive reference.

The authors acknowledge the observation provided, and supportive references have been included in the statement (lines 1,021-1,029). The phrase previously stated as “...using these promising approaches” has been revised to read “...using these approaches.”

Line 961: Review the statement “An important consideration is the possible use of different adjuvants and its relevance on vaccine formulations in further enhancing the immunogenicity for both DNA and RNA vaccines”. This is contradictory to the statement given in lines 473-475: “Additionally, these vaccines do not require live pathogens or adjuvants, reducing safety concerns associated with traditional vaccine platforms”. Which adjuvants can be used for DNA and RNA vaccines?

The authors appreciate the observation provided. ​The statement in lines 473-475 has been revised to read: “Furthermore, while some of these vaccines are formulated with adjuvants, others are capable of eliciting a potent immune response as unadjuvanted formulations.” (lines 441-443). In this context, it is worth noting that DNA and RNA vaccines may utilize adjuvants, including chemical adjuvants (such as alum and CpG) and genetic adjuvants (such as plasmids encoding cytokines like IL-10, IL-12, or IFN-γ) (Grunwald and Ulbert, 2015).

Lone 979: It is mainly produced by CD4+ T cells, but CD8+ T cells. This statement is incomplete.

This sentence has been corrected accordingly (lines 1,066-1,067).

Line 1003-1005: Review the statement: It is known that E2 and NS3 play critical roles in the viral life cycle and immune evasion, making them important targets for vaccine development.

The revised text now properly depicts the implemented alteration (lines 1,147-1,149). The phrase has been revised to read “The E2 and NS3 proteins are essential for the virus life cycle, thereby rendering them important targets for vaccine development”.

Figure 3 legend: Remove repeat words.

This figure legend has been corrected accordingly.

All discussion section: The results discussed from respective studies should also include what animal model (or clinical trial) it refers to.    

The authors appreciate the observation provided.

Other:

Overall, the layout (headings and subheadings) of the manuscript is little difficult to follow for a reader. A clear layout could include listing the structural, non-structural proteins, and a combination of structural + non-structural proteins, and then describing all the vaccination strategies that have been tested for each protein in that category (which could come under different subheadings). In this way both the proteins and the vaccination strategies would be more distinctly laid out. For example, below: 

(A) Structural proteins:

  1. Core
  2. Envelope glycoprotein 1
  3. Envelope glycoprotein 2

(B) Non-structural proteins:

  1. P7
  2. NS2
  3. NS3
  4. NS4A
  5. NS4B
  6. NS5A
  7. NS5B

(C) Combination of Structural and Non-structural proteins 

The authors appreciate the reviewer's suggestions. In this regard, new subheadings have been added to delineate the various proteins targeted by each approach: 

  1. Peptide/protein-based vaccines

3.1 Structural proteins

Core protein

Envelope glycoprotein 1 and 2

Soluble E2 expression using different systems and their formulation with various adjuvants

Antigen presentation by viral vectors

The antigen expression by virus-like particles

3.2 Non-structural proteins

P7

NS3 and NS5B

NS3, NS4A and NS4B

NS2, NS3, NS4A, NS4B, NS5A and NS5B

 3.3 Combination of the structural and non-structural proteins

Viral vector protein expression

E1, E2, NS3 and NS4A

E1, E2, NS3, NS4A, NS4B, NS5A and NS5B

Full-length HCV genome

3.4 Other approaches

Core, E1, E2 and NS3

E1, E2, NS4B, NS5A and NS5B

  1. DNA-based vaccines

4.1 Structural proteins

Core

Envelope glycoprotein 1 and 2

4.2 Non-structural proteins

NS2

NS3

NS3 and NS4A

NS5A

NS3, NS4A, NS4B, NS5A and NS5B

Viral vectors in DNA-based vaccines

NS3, NS4A, NS4B and NS5B

Vaccination of non-human primates

NS3, NS4A, NS4B, NS5A and NS5B 

4.3 Combination of the structural and non-structural proteins

Core, E2, NS3 and NS5B

Core, E1, E2, p7, NS2 and NS3

Core, E1, E2, NS2, NS3, NS4 and NS5

HCV DNA delivered by viral vectors

Core, E1, E2 and NS3

Full-length HCV genome

  1. RNA-based vaccines

Envelope glycoprotein 1 and 2

  1. Human clinical trials

Clinical trials involving viral vectors

NS3 and NS5B

NS3, NS4A, NS4B, NS5A and NS5B

Intramuscular electroporation for HCV vaccine delivery in human clinical trials

NS3 and NS4A

NS3, NS4A, NS4B and NS5A

Line 29: Minor edit required: It is crucial to understand that… 

This sentence has been corrected accordingly (lines 29-30).

Line 1155: Ref 15: Grakoui, A.; Shoukry, N.H.; Woollard, D.J.; Han, J.-H.; Hanson, H.L.; Ghrayeb, J.; Murthy, K.K.; Rice, C.M.; Walker,  C.M. HCV Persistence and Immune Evasion in the Absence of Memory T Cell Help. Science (1979) 2003, 302, 659–662. Please remove year (1979). The correct year of publication is 2003.

This reference has been corrected accordingly.

Round 2

Reviewer 1 Report

Comments and Suggestions for Authors

The authors satisfactorily addressed thè raised points and the manuscript can be accepted.

Reviewer 2 Report

Comments and Suggestions for Authors

Thank you for revising the manuscript based on reviewer comments.

There are no further comments.